# Constraining urban fossil fuel CO₂ emissions in Seoul using combined ground and satellite observations with Bayesian inverse modelling

Sojung Sim[1,2], Sujong Jeong[2,3]

[1]Environmental Planning Institute, Seoul National University, Seoul, 08826, Republic of Korea
[2]Climate Tech Center, Seoul National University, Seoul, 08826, Republic of Korea
[3]Department of Environmental Management, Graduate School of Environmental Studies, Seoul National University, Seoul, 08826, Republic of Korea

*Correspondence to*: Sujong Jeong (sujong@snu.ac.kr)

**Abstract.** Accurate carbon emission estimates are essential for achieving net zero targets by 2050. The Bayesian inverse
method, combined with atmospheric carbon dioxide ($CO_2$) measurements and a transport model, can serve as an independent verification approach to improve accuracy. We developed a Bayesian inverse modelling framework using ground- and space-based measurements and applied it to Seoul to test the framework and constrain its fossil fuel $CO_2$ emissions. By leveraging the high temporal resolution of ground-based in situ observations and the broad spatial coverage of satellite data, we improved the accuracy of emission estimates. Our results indicate the spatiotemporal variability of posterior emissions increased
significantly, enabling us to track $CO_2$ fluctuations and assess the impact of carbon reduction policies over time and space. The mean absolute error between simulated and observed $CO_2$ enhancements decreased by 55%, indicating improved agreement. We investigated the performance of the inverse model through a sensitivity analysis that considered different observational network configurations. Uncertainty reductions varied with the type of observations used: 19.2% when all observations were included, 18.7% using only ground-based sites, and 6–8.4% when using only OCO-2 or OCO-3 satellite
data, highlighting the complementary contributions of ground and space-based measurements. The analysis also showed that assumptions about background concentrations, biogenic fluxes, and prior emission uncertainties can alter posterior results, demonstrating the importance of model configuration. The framework shows strong potential for application in other cities and can support the development of effective climate mitigation policies.

## 25 Graphical abstracts

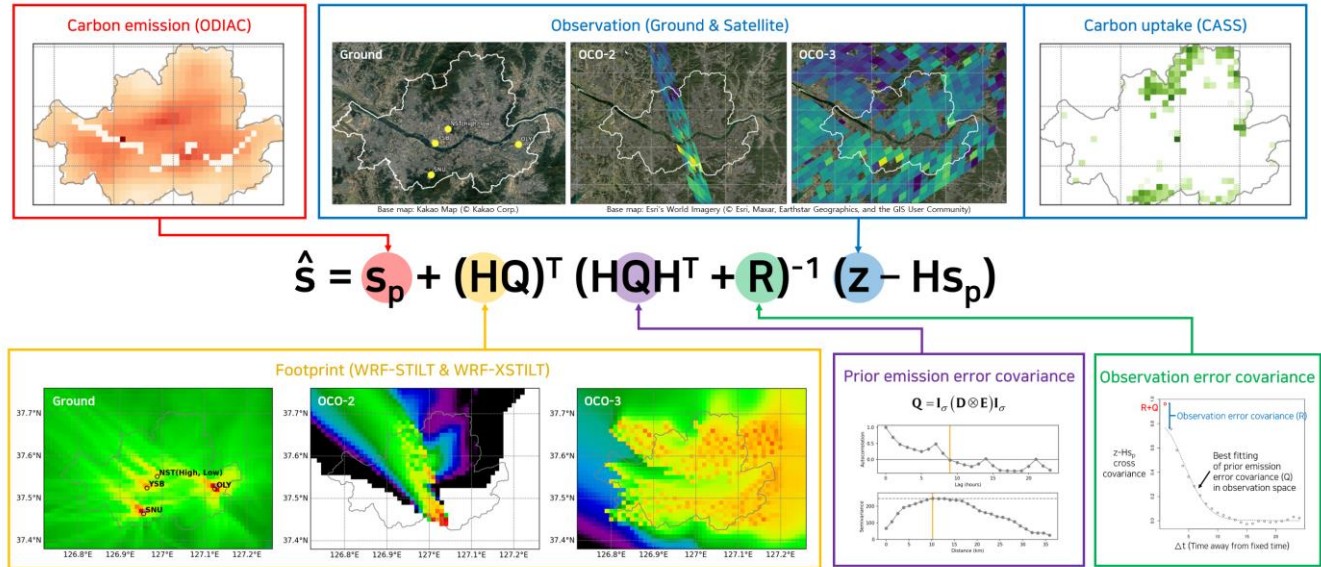

## 1 Introduction

Global carbon emissions from fossil fuel combustion and land-use change are redistributed among the atmosphere, ocean, and
land (Friedlingstein et al., 2022). In the global carbon cycle, these five major components closely interact to maintain balance.
However, anthropogenic emissions of greenhouse gases (GHGs), particularly carbon dioxide ($CO_2$), have dominated since the
industrial era, resulting in rising atmospheric $CO_2$ levels and driving climate change (Friedlingstein et al., 2022). Human-
caused climate change amplifies global surface temperatures and impacts weather extremes such as heatwaves, tropical
cyclones, droughts, and heavy precipitation (IPCC, 2023). To mitigate the adverse effects of climate change worldwide,
international climate agreements like the United Nations Framework Convention on Climate Change, the Kyoto Protocol, and
the Paris Agreement have been implemented (IPCC, 2023). The IPCC 1.5°C Special Report declared that achieving global
net-zero $CO_2$ emissions by 2050 is imperative to limit the increase in global temperature to 1.5 °C above pre-industrial levels
(IPCC, 2018). More than 130 countries have signalled an intention to reduce $CO_2$ emissions to near net-zero by around mid-
century (Robinson and Shine, 2018).
By 2050, 68% of the world's population is projected to reside in urban areas (UN-DESA, 2018). At least 70% of global
anthropogenic $CO_2$ emissions originate from cities (IEA, 2015). Given the concentration of population and $CO_2$ emissions in
cities, they bear significant responsibility for emission reduction and are expected to play a major role in meeting their net-
zero goals. The C40 Climate Leadership Group, comprising around 100 cities worldwide, has pledged to reduce GHG

emissions and developed a science-based approach (C40 Cities, 2022). To support urban emission reduction strategies, the

quality assurance of $CO_2$ emission estimates is required (Gurney et al., 2021; IPCC, 2019). Accurate emission estimates can provide detailed guidance to establish a baseline for prioritizing climate action and assessing policy progress over time (Mueller et al., 2021).

Various efforts are underway to estimate $CO_2$ emissions through a bottom-up approach. This approach calculates anthropogenic $CO_2$ emissions based on socio-economic databases (e.g., energy consumption, housing statistics, and road

networks) and emission factors. The Global Protocol for Community-Scale Greenhouse Gas Emission Inventories (GPC) provides a standardized framework adopted by major city networks such as C40 Cities (WRI et al., 2014). It offers consistent accounting methods for different emission sectors within urban boundaries, enabling cities worldwide to compile self-reported emission inventories. However, these inventories are typically reported at an annual scale, resulting in low spatial and temporal resolution. Because of limited access to activity data and unknown emission factors within urban areas, detailed $CO_2$ emission

estimates with high spatial and temporal resolution have been developed only for certain cities, such as Indianapolis (Gurney et al., 2012), Los Angeles (Feng et al., 2016; Gurney et al., 2019b), Salt Lake City (Patarasuk et al., 2016), and New York (Gately et al., 2015). Other types of emission data products, including CDIAC (Carbon Dioxide Information and Analysis Center), EDGAR (Emissions Database for Global Atmospheric Research), and ODIAC (Open-Data Inventory for Anthropogenic Carbon Dioxide), have been devised to downscale global/national total emission estimates using proxies such

as nighttime lights and population. However, large discrepancies have been reported between bottom-up and GPC-based inventories and downscaled $CO_2$ datasets at the urban scale. These differences arise from variations in emission sources (e.g., large point sources and road traffic), outdated local-specific emission factors, and inconsistencies in spatial or temporal coverage (Ahn et al., 2023; Gurney et al., 2019a; Palermo et al., 2024). Such uncertainties limit the establishment of a $CO_2$ emissions baseline and assessing mitigation outcomes at the city levels.

A complementary and independent approach to verify these $CO_2$ emission estimates is deemed necessary (IPCC, 2019). In the top-down approach, emission estimates can be constrained via real-time $CO_2$ measurements and atmospheric transport models. Consequently, combining bottom-up and top-down estimates has been explored using a Bayesian inversion approach for accurate $CO_2$ emission estimation. Bayesian inversion approach has been used in recent studies to optimize existing bottom-up estimates over Salt Lake City (Kunik et al., 2019; Mallia et al., 2020), Paris (Lian et al., 2022; Nalini et al., 2022), Los

Angeles (Ye et al., 2020), and Tokyo (Ohyama et al., 2023; Pisso et al., 2019). They obtained optimal $CO_2$ emissions with uncertainty reductions of 39.32% (Kunik et al., 2019), 27.7% (Mallia et al., 2020), 8–10% (Lian et al., 2022), 2–10% (Nalini et al., 2022), ~50% (Ohyama et al., 2023), and 20.09% (Pisso et al., 2019) compared to prior emissions.

Previous studies that performed inverse modelling for urban areas have primarily relied on a single type of observation to constrain emissions, most commonly ground-based in situ $CO_2$ measurements (Breón et al., 2015; Göckede et al., 2010;

Lauvaux et al., 2016; Lian et al., 2023, 2022; Mallia et al., 2020; McKain et al., 2012; Nalini et al., 2022; Sargent et al., 2018; Staufer et al., 2016). Some studies have instead used ground-based Fourier Transform Infrared (FTIR) column-averaged $CO_2$ ($XCO_2$) observations (Hedelius et al., 2018; Ohyama et al., 2023), airborne observations (Lopez-Coto et al., 2020; Pitt et al.,

2022), or satellite observations (Hamilton et al., 2024; Kaminski et al., 2022; Roten et al., 2023; Wu et al., 2018; Ye et al., 2020). However, studies that combine multiple observation types to leverage their complementary strengths remain rare. Although Pisso et al. (2019) integrated in situ airborne and ground-based observations to assess Lagrangian inverse modelling, and Che et al. (2024) combined ground-based FTIR and satellite data to estimate $CO_2$ emissions, these studies focused on either near-surface $CO_2$ concentrations or vertical $CO_2$ profiles, rather than incorporating both perspectives. In this study, we integrate ground-based in situ $CO_2$ observations, which provide detailed information on surface emissions and uptake with high temporal resolution, and satellite observations, which offer broad spatial coverage and capture the total atmospheric $CO_2$ column. By combining these two complementary datasets, we simultaneously account for surface $CO_2$ fluxes and their impact on the vertical distribution of atmospheric $CO_2$. To our knowledge, this is the first inverse modelling study to fully utilize surface and column-integrated $CO_2$ measurements, providing a more comprehensive constraint on urban $CO_2$ emissions.

Seoul is a megacity with a population of approximately 10 million, which accounts for 18% of South Korea's total population in 2022 (KOSIS, 2023). The population and infrastructure in Seoul are densely concentrated, making it more susceptible to severe damage from climate change than other regions. It also has one of the highest carbon emissions among the 13,000 cities worldwide (Moran et al., 2018). Seoul has participated in the C40 Climate Leadership Group since 2006, and in 2020, it announced the '2050 GHGs Reduction Promotion Plan' to achieve a net-zero emissions goal (Seoul Metropolitan Government, 2021). Within Seoul, a comprehensive $CO_2$ monitoring network has been established, encompassing numerous stationary monitoring sites and mobile platforms to understand the urban carbon cycle (Park et al., 2020; Sim et al., 2020). Given its dense population, concentrated emissions, and extensive measurement networks, Seoul can be an optimal testbed city for studies to verify $CO_2$ emission estimates and assess the effectiveness of the $CO_2$ monitoring network.

In this study, we used a Bayesian inverse model and ground- and space-based measurements to improve the accuracy of $CO_2$ emission estimates over Seoul. We developed a high-resolution Bayesian inverse modelling framework with a spatial resolution of 0.01° (approximately 1 km) and a temporal resolution of 1 h, incorporating anthropogenic $CO_2$ emissions, biogenic $CO_2$ fluxes, atmospheric $CO_2$ measurements, a Lagrangian transport model, and error covariances of both prior emissions and observations. We then estimated the optimal spatiotemporal distribution of $CO_2$ emissions over Seoul for December 2021, verifying existing emission data. Additionally, we evaluated the effectiveness of the inversion by comparing observed and simulated $CO_2$ enhancements using prior and posterior emissions. Finally, we conducted sensitivity tests on different observational datasets and input assumptions to assess their impact on emission estimates.

## 2 Data and methods

### 2.1 Bayesian inverse method

In this study, data assimilation is employed to estimate optimal (posterior) $CO_2$ emissions close to the true emissions. Data assimilation in $CO_2$ estimation optimally combines information from atmospheric $CO_2$ observations with a transport model

and prior $CO_2$ emissions to produce accurate posterior estimates of $CO_2$ emissions. Posterior $CO_2$ emissions are derived through the minimization of the cost function (Enting, 2002; Tarantola, 1987) defined as follows:

$$L_s = \frac{1}{2}(z - Hs)^T R^{-1}(z - Hs) + \frac{1}{2}(s - s_p)^T Q^{-1}(s - s_p) \tag{1}$$

Where $z$ is a vector of observed $CO_2$ enhancements, $H$ is the Jacobian matrix of footprint values from the atmospheric transport model, $s$ is a vector of the unknown true $CO_2$ emissions, $R$ is the covariance of observational errors, $s_p$ is a state vector of prior $CO_2$ emissions, and $Q$ is the covariance of prior emission errors. The solution obtained by minimizing the cost function defined in Eq. (1) yields the optimized posterior $CO_2$ emission estimates ($\hat{s}$), expressed as:

$$\hat{s} = s_p + (HQ)^T (HQH^T + R)^{-1}(z - Hs_p) \tag{2}$$

The posterior uncertainty covariance ($V_{\hat{s}}$) can be expressed as:

$$V_{\hat{s}} = Q - (HQ)^T (HQH^T + R)^{-1}(HQ) \tag{3}$$

Using the posterior uncertainty covariance obtained from Eq. (3), the reduction in uncertainty resulting from the constraints on emissions can be quantified. The uncertatinty reduction (UR) is defined as:

$$UR = \frac{\sqrt{Q_{tot}} - \sqrt{V_{\hat{s}\_tot}}}{\sqrt{Q_{tot}}} \times 100\% \tag{4}$$

Here, $Q_{tot}$ represents the domain- and time-averaged covariance of prior emission errors and $V_{\hat{s}\_tot}$ represents the domain- and time-averaged covariance of posterior emission errors.

In this study, we assessed the validity of posterior emissions resulting from the inverse model by calculating the reduced chi-squared value ($\chi_r^2$) following Tarantola (1987). It is computed using the equation:

$$\chi_r^2 = \frac{1}{\nu}[(z - H\hat{s})^T R^{-1}(z - H\hat{s}) + (\hat{s} - s_p)^T Q^{-1}(\hat{s} - s_p)] \tag{5}$$

Where the squared data residual ($z - H\hat{s}$) and emissions residual ($\hat{s} - s_p$) from the inversion are normalized by their respective variance matrices, $R$ and $Q$. The residuals are expected to follow a chi-squared distribution with $\nu$ degrees of freedom, which in this study corresponds to the number of observations. The closer the reduced chi-squared value is to 1, the more accurately the prescribed errors of observations and prior emissions are set, leading to a more reliable estimation of posterior emissions.

## 2.2 Observations

A set of measurements from ground-based and satellite observations, all collected during December 2021, was selected to obtain atmospheric $CO_2$ concentrations in Seoul. The year 2021 was chosen because it was the first year when $CO_2$ data from all ground-based observation sites in Seoul became available. December was further selected to minimize the influence of biogenic activity on the inversion results and because both OCO-2 and OCO-3 satellites passed over Seoul during this period, enabling consistent integration of ground-based and satellite observations. Ground-based observations provide continuous, real-time measurements of atmospheric $CO_2$ at specific locations, whereas satellite observations offer broader spatial coverage that complements the ground network.

140

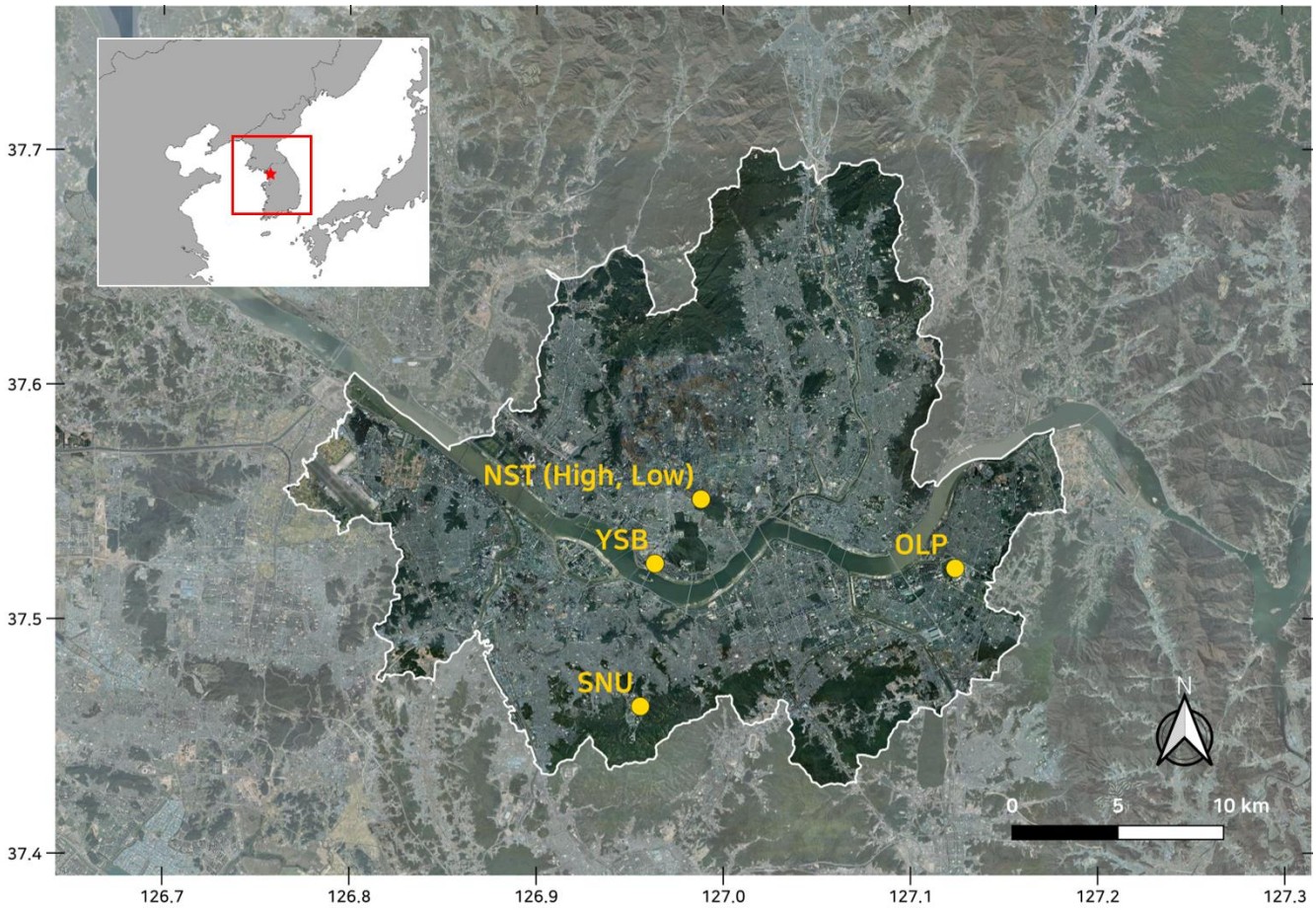

**Figure 1: Map of Seoul with the locations of CO$_2$ ground observation sites: Namsan Seoul Tower-High (NST$_H$), Namsan Seoul Tower-Low (NST$_L$), Olympic Park (OLP), Seoul National University (SNU), and Yongsan Building (YSB). Observation sites are denoted with yellow points. The city's elevation generally ranges from 20 to 80 m above sea level, with higher terrain in the northern and southern mountainous areas. The background map is sourced from Kakao Map (© Kakao Corp.). The red box in the inset figure at the upper left indicates the background domain used for satellite-based observations.**

For the ground measurements, we used observed CO$_2$ concentrations from five different sites in Seoul (Park et al., 2021): Namsan Seoul Tower-High (NST$_H$), Namsan Seoul Tower-Low (NST$_L$), Olympic Park (OLP), Seoul National University (SNU), and Yongsan Building (YSB), as shown in Fig. 1. The instrument inlet heights of NST$_H$, NST$_L$, OLP, SNU, and YSB, combined with the site altitude, are 420, 265, 27, 173, and 113 m above sea level, respectively. The observation instruments installed at NST$_L$ and OLP are PICARRO's G2301, whereas those installed at NST$_H$, SNU, and YSB are LICOR's LI-850. We utilized only daytime data (10:00–16:00 KST) for the inverse modelling to minimize the impact of model biases in the planetary boundary layer (PBL) height. The CO$_2$ concentrations measured at each ground observation site exhibit different patterns of variation because of differences in altitudes and surrounding environments (Fig. 2a). The average daytime CO$_2$

concentrations at $NST_H$, $NST_L$, OLP, SNU, and YSB were $453.7 \pm 19.6$, $460.1 \pm 27.1$, $460.2 \pm 26.7$, $461.5 \pm 26.6$, and $466.9 \pm 31.4$ ppm, respectively (Fig. 2b). The lowest average concentration at $NST_H$ is attributed to its location at the top of the tall tower, whereas the highest average concentration at YSB is because of its location in a commercial area with high vehicle traffic. Most observation sites exhibited a typical diurnal pattern of $CO_2$ concentrations (Fig. 2c), characterized by an increase

in the morning as emissions rise (Fig. S4b), followed by a pronounced decrease during the daytime when the PBL height increases and atmospheric mixing becomes more active. At night, although emissions are relatively lower, the shallow PBL leads to an accumulation of $CO_2$ near the surface, resulting in higher concentrations. However, at the $NST_H$ site, $CO_2$ concentrations tended to decrease at night, showing an opposite pattern compared to the other sites. This is because the observation inlet at $NST_H$ is often located above the nocturnal PBL, allowing it to be influenced by air masses transported from

outside Seoul (Park et al., 2022).

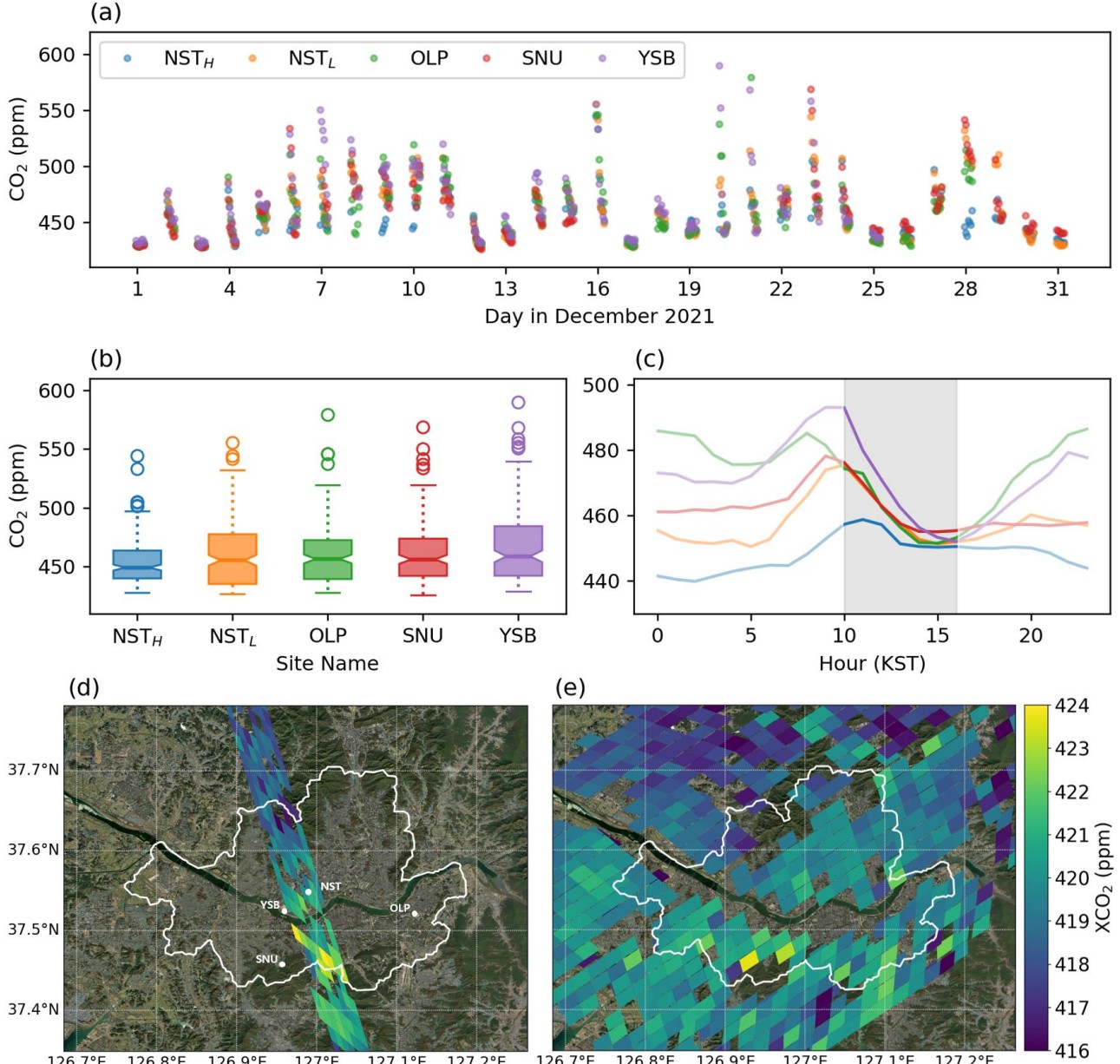

**Figure 2: Observation data used in the Bayesian inverse modelling framework over Seoul during December 2021. (a) Time series of daytime (10:00–16:00 KST) CO₂ concentrations measured at five ground sites. (b) Box plots of daytime CO₂ concentrations from the ground sites. (c) Average diurnal cycles at each site, with gray shaded regions indicating the daytime period. Spatial distributions of XCO₂ measured by (d) OCO-2 on December 4, 2021, at 13:00 KST and (e) OCO-3 on December 5, 2021, at 11:00 KST. White circles in panel (d) indicate the locations of the ground observaton sites. The background maps in (d) and (e) are based on Esri's World Imagery (© Esri, Maxar, Earthstar Geographics, and the GIS User Community).**

We used satellite measurements from the Orbiting Carbon Observatory-2 (OCO-2) and Orbiting Carbon Observatory-3 (OCO-3) missions. OCO-2 and OCO-3, launched on July 2, 2014, and May 4, 2019, respectively, measure the dry-air column-averaged mole fraction of $CO_2$ ($XCO_2$). Both satellites provide high-precision spaceborne observations at a fine spatial resolution, making them highly suitable for urban-scale inverse modelling. In particular, OCO-3 offers the Snapshot Area Mapping mode, specifically designed to detect anthropogenic $CO_2$ emissions and spatial gradients in densely populated urban environments (Kiel et al., 2021). These capabilities make both satellites valuable for constraining urban $CO_2$ emissions. Over Seoul, valid OCO-2 and OCO-3 soundings have been intermittently obtained, with data availability varying by month. In 2021, no valid observations were available in February, March, April, August, and September. Only OCO-2 passed over Seoul in July and November, while only OCO-3 provided soundings in January, May, and June. Both satellites observed Seoul in October and December. Among these, December 2021 offered broader spatial coverage and higher-quality retrievals, and was therefore selected for this study. On December 4 and 5, 2021, the OCO-2 and OCO-3 satellites passed over Seoul at 13:00 and 11:00 KST, respectively, yielding 60 and 167 soundings (Figs. 2d and 2e). Only good-quality data were considered when processing the $XCO_2$ data for observational input in inverse modelling. In both datasets, the $XCO_2$ values were higher in the southern part of Seoul compared to the northern part. This spatial gradient likely reflects differences in topography and emission patterns. The northern boundary of Seoul is mountainous, resulting in lower $XCO_2$ levels due to limited anthropogenic activity, whereas the southern part contains densely populated residential and commercial zones with higher emission intensities.

The vector of observed $CO_2$ enhancements, denoted as $\mathbf{z}$ in Eq. (1), represents the $\triangle CO_2$ affected by nearby emission sources. Because we aim to optimize Seoul's $CO_2$ emissions using atmospheric observations, we must calculate $\triangle CO_2$ influenced only by anthropogenic emissions within Seoul, excluding the effects of biogenic fluxes and background. To obtain the $\triangle CO_2$, the vegetation-affected and background concentrations must be subtracted from the observed $CO_2$ concentrations. The calculation method for the vegetation-affected concentration is described in Sect. 2.4.

To isolate $\triangle CO_2$ from observed $CO_2$ concentrations at ground observation sites, we determined suitable background representations for the Seoul region. Previous studies have proposed several approaches to estimate background concentrations, including the use of data from background sites, upwind sites, or the observation sites themselves. When background sites are utilized, various data selection strategies have been applied, such as using hourly measurements from high-altitude stations (Nalini et al., 2022), daily minimum values (Fasoli et al., 2018; Zhao et al., 2009), the daily 5[th] percentile (Ohyama et al., 2023), curve-fitted data (Thoning et al., 1989), or data processed with the Robust Extraction of Baseline Signal algorithm (Ruckstuhl et al., 2012). In approaches considering wind direction, background concentrations have been represented by hourly mixing ratios (Lian et al., 2022; Nalini et al., 2022) or by two-day moving averages of $\triangle CO_2$ from an upwind site (McKain et al., 2012). When background or upwind stations are unavailable, background concentrations can be derived directly from site-specific observations using statistical methods, such as the 24-hour moving 5[th] percentile (Chandra et al., 2016; Gamage et al., 2020) or the 3-day moving 5[th] percentile (Ammoura et al., 2014).

For the background representation for ground-based observations in Seoul, we calculated observed $\triangle CO_2$ at each site using three approaches: 1) the daily 5th percentile at $NST_H$, a high-altitude urban background site (denoted as GM1); 2) the 24-hour moving 5th percentile (GM2); and 3) the 3-day moving 5th percentile (GM3), both derived from data at each site. Among these methods, GM2 yielded the lowest mean absolute error (MAE) between observed and modelled $\triangle CO_2$ from posterior emissions and was therefore selected as the reference configuration. The sensitivity of inversion results to different ground background estimation methods is discussed in Section 3.3.2.

To define the background for satellite-based observations, various approaches have also been proposed in previous studies, ranging from simple statistical to geographic and model-based methods, depending on study objectives. Hakkarainen et al. (2016) and Silva and Arellano (2017) employed simple statistical approaches, defining the daily median of all observations within a broad domain or the mean minus one standard deviation of all observations within the target urban area, respectively. However, these domain-wide statistical methods may yield background concentrations that represent a mixture of regions with distinct characteristics. To address this limitation, several studies incorporated geographic information to identify rural regions and used observations from these regions to define background levels. For instance, Kort et al. (2013) and Schneising et al. (2013) utilized data over rural or desert regions with relatively low emissions, while Labzovskii et al. (2019) and Park et al. (2021) defined a ~500,000 km$^2$ background box centered on each city and used the daily median $XCO_2$ values over rural areas, identified using MODIS land cover data, as the background concentration. Model-based approaches have also been widely adopted. Janardanan et al. (2016) used an atmospheric transport model and flux data to identify grid cells where the simulated anthropogenic signal was below a specified threshold, averaging the corresponding concentrations to define the background. Wu et al. (2018) compared trajectory-endpoint and overpass-specific background methods using a transport model and found that the overpass-specific method was most suitable for capturing local-scale $XCO_2$ anomalies. They also reported that reducing the spatial domain from $4° \times 4°$ to $2° \times 2°$ in the Silva and Arellano (2017) method yielded results comparable to those from the overpass-specific approach in both variability and magnitude. Similarly, Hamilton et al. (2024) used a transport model to identify upwind regions within the study domain and defined the mean $XCO_2$ over those regions as the background concentration.

Among these various satellite background estimation methods, we selected three representative approaches for this study: 1) a simple statistical method (SM1), following Silva and Arellano (2017); 2) a geographic method (SM2), following Labzovskii et al. (2019) and Park et al. (2021); and 3) a model-based upwind method (SM3), following Hamilton et al. (2024). In SM1, we adopted a $2° \times 2°$ domain centered on Seoul, modifying the original approach based on the recommendation of Wu et al. (2018), and defined the background as the mean minus one standard deviation of all satellite observations within the domain. In SM2, the background was calculated as the daily median $XCO_2$ over non-urban areas within a ~500,000 km² background box centered on Seoul (Fig. 1). Instead of MODIS land cover data used in previous studies, we utilized the land cover map from the Environmental Geographic Information Service (EGIS) of the Ministry of Environment Korea, which more accurately represents domestic geographic conditions (see Text S1 and Fig. S1 in the supplementary material). For SM3, we used a transport model (Section 2.3) to identify upwind regions and defined the mean $XCO_2$ observed in those regions as the

background concentration. Each of the three methods was applied separately to OCO-2 and OCO-3 observations, followed by inverse modelling for sensitivity analyses (Section 3.3.2). Among these, SM2 produced the lowest MAE between observed and modelled $\triangle CO_2$ from posterior emissions and was therefore selected as the reference configuration for this study.


## 2.3 Atmospheric Transport

The Weather Research and Forecasting model with the (X-)Stochastic Time-Inverted Lagrangian Transport (WRF-(X)STILT) was used to derive the Jacobian matrix of footprint values ($H$) at a fine spatial resolution (e.g., 0.01°). We employed the STILT model (Fasoli et al., 2018; Lin et al., 2003) for ground-based observations and X-STILT (Wu et al., 2018) for satellite

observations, both driven by meteorological fields from WRF model version 3.9.1 (Skamarock and Klemp, 2008). WRF-(X)STILT is an effective tool for simulating realistic atmospheric transport using a Lagrangian particle dispersion model within the planetary boundary layer (Nehrkorn et al., 2010). Previous studies have widely used WRF-(X)STILT as an atmospheric transport model for applying GHG inverse modelling in urban areas (Kunik et al., 2019; McKain et al., 2012; Ohyama et al., 2023; Sargent et al., 2018; Wu et al., 2018; Zhao et al., 2009).

The model releases backward 3D virtual air particle trajectories with stochastically turbulent dispersion from the observation location (receptor) to potential source regions that influence the receptor. It then counts the dispersed air particles (footprints) in each grid. Footprints quantify the sensitivity of the observation to upstream source regions. They can be regarded as the average contribution of the surface flux at the receptor, as they represent how densely and how long the air particles lingered backward in time within each discretized volume of the upwind source regions. In Bayesian inverse modelling, the footprint

acts as an operator, connecting individual $CO_2$ observations (unit: ppm) and gridded fluxes (unit: $\mu mol/(m^2\,s)$). Using footprints representing concentration per unit flux allows direct comparison between $CO_2$ emissions and atmospheric $CO_2$ enhancements. To ensure the reliability of the meteorological inputs used for the inversion, the modelled wind fields from the WRF simulation were evaluated against observational data from the Automated Surface Observing System (ASOS) and Automatic Weather Stations (AWS) operated by the Korea Meteorological Administration. Several sensitivity cases were tested to assess the

effects of meteorological inputs, land-use data, and grid nudging on model performance. The mean wind speed bias in Seoul decreased from 2.38 m/s (WRF default configuration) to 0.96 m/s (ERA5/EGIS with grid nudging) at the ASOS site and from 2.95 m/s to 1.46 m/s on average across 28 AWS stations. Based on these results, the configuration using ERA5 reanalysis data, EGIS land-use information, and grid nudging yielded the most accurate meteorological fields and was therefore adopted for the inversion simulations. Detailed evaluation results are provided in Text S1 of the supplementary material.

For the STILT simulations, one thousand air particles were released from each observation site at the height corresponding to the measurement inlet above ground level (a.g.l.) and tracked backward in time for 24 h (Fig. 3a). A 24-hour backward period was sufficient for particles to travel beyond the innermost WRF domain (Domain 3; Fig. S2), thereby capturing the full regional influence on the observations. The STILT model was run hourly for each observation site, producing hourly footprints that represent the sensitivity of the measured $CO_2$ mole fraction to surface fluxes at each backward time step. The footprints were

computed for an effective mixing depth below half the height of the boundary layer, considering time-varying vertical mixing depths that depend on the receptor location and meteorological conditions (Fasoli et al., 2018). The resulting footprints were vertically averaged to provide a representative surface influence field.

X-STILT extends STILT by explicitly accounting for satellite vertical profiles in the footprint calculation. In the X-STILT simulations, one thousand air particles were released from each column level for OCO-2 and OCO-3 soundings and tracked

backward for 24 h (Figs. 3b and 3c). The column receptors consisted of 37 vertical levels with 100 m spacing up to 3000 m a.g.l. and 500 m spacing up to 6000 m a.g.l. Thereafter, no particles were released above 6000 m a.g.l. to reduce computational cost, as contributions from higher altitudes to total column sensitivities are negligible. For each sounding, the modelled column sensitivity was multiplied by the corresponding column-averaging kernel and pressure-weight vector to represent the $XCO_2$ signal. Details and equations for the footprint calculations in STILT and X-STILT are provided in Text S2 of the supplementary

material.

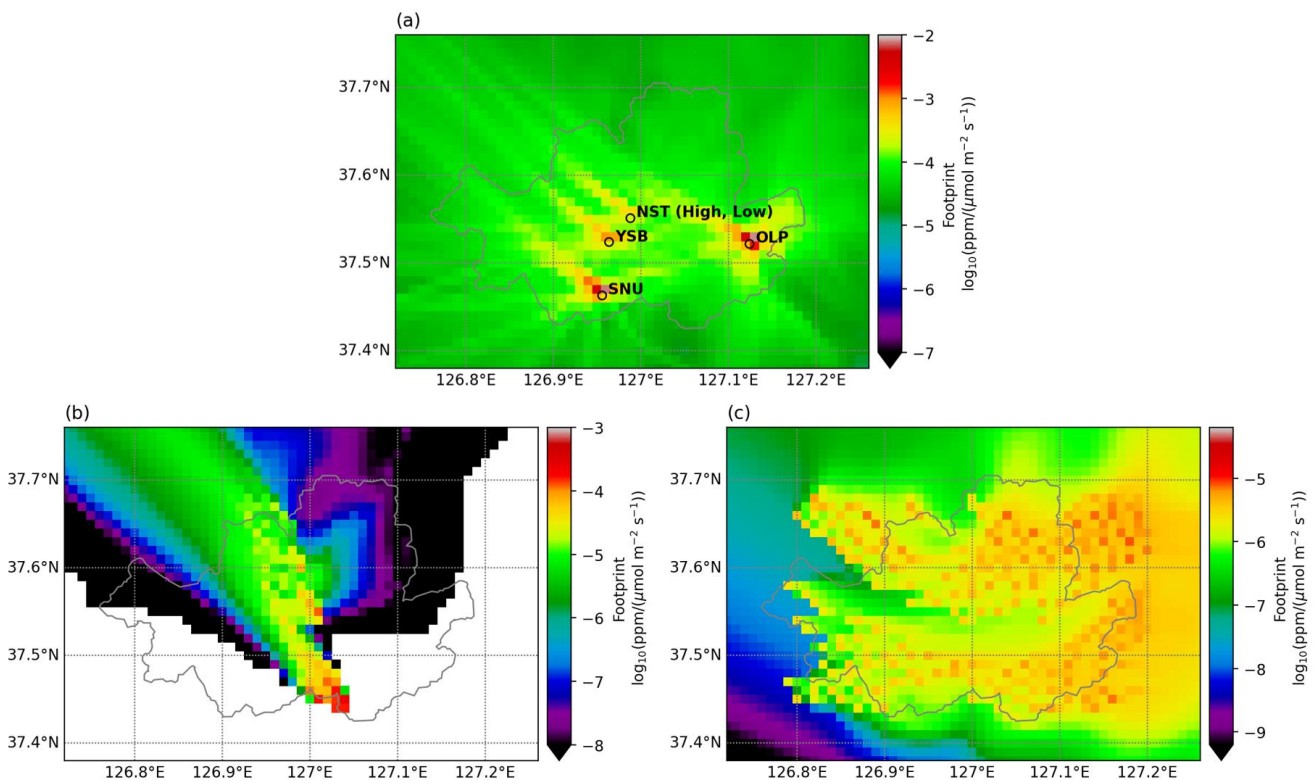

**Figure 3: Footprint averages on a log$_{10}$ scale to upwind source regions of (a) daytime-only ground observations (10:00–16:00 KST) during December 2021, (b) OCO-2 satellite data on December 4, 2021, at 13:00 KST, and (c) OCO-3 satellite data on December 5,**
**2021, at 11:00 KST across the Seoul domain. Note that the footprint ranges differ in panels (a), (b), and (c) for visualization purposes.**

## 2.4 CO$_2$ emissions & biogenic fluxes

### 2.4.1 Anthropogenic CO$_2$ emissions

We used anthropogenic CO$_2$ emissions data from ODIAC version 2022 as prior emissions (Oda et al., 2018; Oda and
Maksyutov, 2011). ODIAC provides global fossil fuel CO$_2$ emission estimates at a high spatial resolution of $1 \times 1$ km$^2$, using
power plant profiles and space-based nighttime light data (Oda et al., 2018). ODIAC is based on the CDIAC national emissions
estimates, which categorize emissions by fuel type—liquid, gas, solid fuel, cement, gas flare, and international bunker. The
ODIAC dataset has been widely used in various research areas, such as urban emission evaluation, monitoring network design
experiments, and the inverse estimation of CO$_2$ emissions (Che et al., 2024; Crowell et al., 2019; Fasoli et al., 2018; Hedelius
et al., 2018; Kunik et al., 2019; Lauvaux et al., 2016; Lian et al., 2023, 2022; Mallia et al., 2020; Ohyama et al., 2023; Sim et
al., 2023; Wu et al., 2018; Ye et al., 2020).

ODIAC is based on downscaling bottom-up CO$_2$ emission estimates using spatial proxies. Global geolocation information and
power plant magnitudes are sourced from the Carbon Monitoring and Action database. However, this database occasionally
misplaces point sources, necessitating manual correction (Kunik et al., 2019; Ohyama et al., 2023). For instance, the Korea
Central Power Corporation in Seoul, located in the western part of the city, is inaccurately positioned in the ODIAC data. To
correct this spatial discrepancy, we customized the ODIAC data, as shown in Fig. S3 of the supplementary material. We
manually relocated the misaligned point source to match the power plant location. Additionally, for the grid cell with
significantly high CO$_2$ emissions mistakenly identified as a point source, we replaced the emission value with the average of
the surrounding eight grid cells.

Because the temporal resolution of ODIAC is monthly, it is necessary to refine the data to achieve hourly CO$_2$ emission
estimates. We applied weekly and diurnal temporal scaling factors (Nassar et al., 2013) specific to Seoul to adjust the monthly
ODIAC emission data, as shown in Fig. S4 in the supplementary material. The weekly scaling factors for Seoul remain
consistent on weekdays, with values exceeding 1, but decrease over the weekends (Fig. S4a). The diurnal scaling factors exhibit
typical daily emission patterns, increasing in the morning, peaking in the afternoon, and decreasing in the evening, reflecting
both energy usage for heating and underlying sociodemographic activity patterns (Fig. S4b). Figure S4c shows the hourly CO$_2$
emissions time series after applying these temporal scaling factors. The emissions, after spatial and temporal pre-processing
for December 2021, serve as the state vector for the prior CO$_2$ emissions, denoted as '$s_p$' (Fig. 4a).

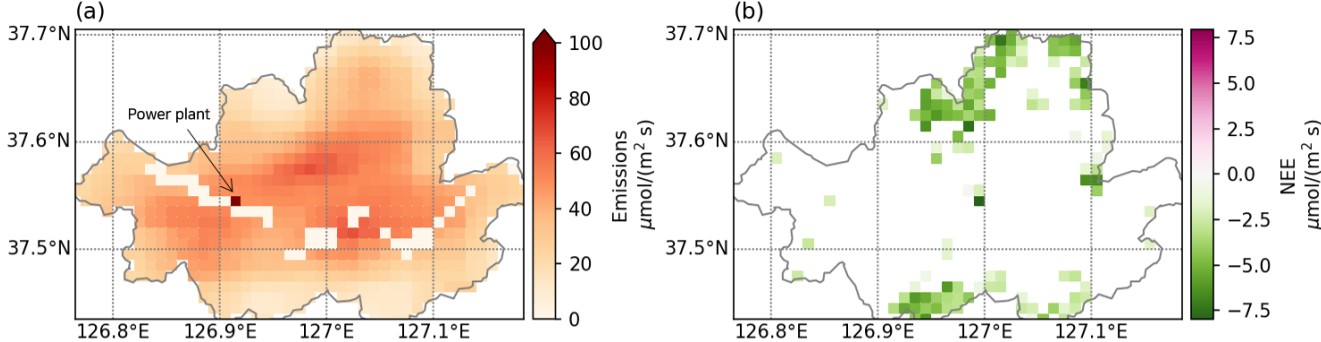

Figure 4: Averaged $CO_2$ fluxes of (a) prior emissions and (b) net biogenic ecosystem exchange over Seoul for daytime (10:00–16:00 KST) in December 2021. In the original ODIAC dataset, the power plant location was represented near the city center (see Fig. S3); in this study, it was corrected to its actual geographical position.

### 2.4.2 Biogenic $CO_2$ fluxes

In the urban carbon cycle, atmospheric $CO_2$ is mainly influenced by emissions from fossil fuel combustion and vegetation carbon uptake. Given that Seoul has forests covering 25.3% of its total area (Korea Forest Service, 2021), the impact of biogenic $CO_2$ fluxes cannot be ignored. To account for the influence of biogenic $CO_2$ on the observed concentration, we incorporated biogenic $CO_2$ fluxes estimated by a data-based model known as CASS (Carbon Simulator from Space). CASS generates terrestrial carbon flux data from vegetation using information such as air temperature, relative humidity, photosynthetically active radiation, enhanced vegetation index, and land surface water index. CASS employs the random forest method to determine optimal coefficients for each region and applies them to the estimation of carbon uptake. We utilized hourly net ecosystem exchange (NEE) data, which were resampled from a 250-meter resolution to 0.01°, as the biogenic $CO_2$ flux data within Seoul (Fig. 4b). $CO_2$ uptake can be observed in grids where Seoul's mountains and parks are located during the daytime. We obtained the vegetation-affected concentration by multiplying the footprints from the atmospheric transport model by the gridded biogenic $CO_2$ flux data using Eq. (S2) or (S4).

### 2.5 Prior error covariance

The prior error covariance matrix ($\boldsymbol{Q}$) is derived from both the variance in prior emissions uncertainty ($\boldsymbol{\sigma}$) and the temporal and spatial covariances ($\boldsymbol{D}$ and $\boldsymbol{E}$). We construct the prior error covariance matrix as follows:

$$\boldsymbol{Q} = \boldsymbol{I}_\sigma(\boldsymbol{D} \otimes \boldsymbol{E})\boldsymbol{I}_\sigma \tag{6}$$

Where $\boldsymbol{I}_\sigma$ is a diagonal matrix whose elements represent the uncertainty of prior emissions. Instead of directly constructing the full $\boldsymbol{Q}$ matrix, the temporal and spatial error covariance matrices are combined using a Kronecker product ($\otimes$) to reduce computational costs, particularly when dealing with large emission state vectors (Yadav and Michalak, 2013).

In previous studies on urban inverse modelling, three methods for estimating prior emissions uncertainty were mainly identified. Most studies assumed a relative uncertainty for prior estimates, such as 30% uncertainty for each emission source in Central California (Zhao et al., 2009), 15% for large point sources and 85% for the rest in Tokyo (Ohyama et al., 2023), 20% for Los Angeles and 40% for Riyadh and Cairo (Ye et al., 2020), 20% (Lian et al., 2022), and 60% (Nalini et al., 2022) for Paris. Kunik et al. (2019) and Mallia et al. (2020) defined the uncertainty of prior emissions as the difference between the prior and true emission estimates (e.g., Hestia). Another approach to estimating prior emissions uncertainty is inter-comparison with different inventories (Sargent et al., 2018; Wu et al., 2018). In this study, we assumed a relative uncertainty of 15% for large point sources and 100% for the rest of Seoul, similar to the approach used for Tokyo in Ohyama et al. (2023). For grids with prior emissions of 0 $\mu$mol m$^{-2}$ s$^{-1}$, such as rivers, a minimum uncertainty value of 1 $\mu$mol m$^{-2}$ s$^{-1}$ was assigned, following the method of Kunik et al. (2019).

The temporal and spatial covariance matrices are defined using exponential decay equations:

$$\boldsymbol{D} = \exp\left(-\frac{X_\tau}{l_\tau}\right) \tag{7}$$

$$\boldsymbol{E} = \exp\left(-\frac{X_s}{l_s}\right) \tag{8}$$

The temporal covariance is computed based on lag-times ($X_\tau$) between time steps, divided by temporal correlation range parameters ($l_\tau$), where $l_\tau$ represents the time over which the correlation decays significantly. Similarly, the spatial covariance is calculated using separation distances ($X_s$) between grid cells, divided by spatial correlation range parameters ($l_s$), where $l_s$ indicates the distance over which the correlation decays significantly. Previous studies have shown that suitable spatiotemporal correlation parameters vary by city. For example, in Salt Lake City, the temporal and spatial correlations were determined to be 2 d and 6 km, respectively (Kunik et al., 2019), whereas in Tokyo, the parameters were 0 d and 10 km (Ohyama et al., 2023). We performed a lagged autocorrelation function and variogram analysis to determine the optimal temporal and spatial correlation lengths for Seoul, respectively (Fig. S5). Based on the results, the optimal temporal and spatial correlation range parameters for the inversion over Seoul were 9 h and 10 km, respectively.

## 2.6 Observational error covariance

We estimated the observational error covariance ($\boldsymbol{R}$) using the departure-based diagnostics, commonly known as the Hollingsworth/Lönnberg method (Hollingsworth and Lönnberg, 1986; Lönnberg and Hollingsworth, 1986; Rutherford, 1972). Here, observation error includes uncertainties arising from the instrument, transport model, representation, background inflow (boundary conditions), and biogenic fluxes. The departure (or innovation) is defined as $\boldsymbol{z} - \boldsymbol{H}\boldsymbol{s_p}$, representing the difference between the observed $\triangle CO_2$ and the values simulated by WRF-(X)STILT with prior emissions, following Eqs. (S2) and (S4). Because the standard deviation of departures reflects the combined effects of observation and prior emissions errors, we separate their contributions based on certain assumptions. The Hollingsworth/Lönnberg method assumes that prior emissions

errors exhibit spatial correlation, whereas observation errors are spatially uncorrelated (Bormann et al., 2009). Additionally, prior emissions and observation errors are considered independent.

To estimate observation errors, we first compute the covariance of departure pairs as a function of separation distance. A function is then fitted to the covariance values at various distances, excluding the value at zero separation, and extrapolated to estimate the covariance at zero distance. At zero separation, the total variance is decomposed into a spatially correlated component (representing prior emissions error) and an uncorrelated component (representing observation error). Based on these assumptions, the value of the fit gives the prior emissions error in observation space at zero separation, and the observation error is determined by subtracting this value from the total covariance at zero distance.

In this study, we applied the Hollingsworth/Lönnberg method to estimate the observational error covariance for both ground-based and satellite observations. We first divided the observation vector into subsets: $NST_H$, $NST_L$, OLP, SNU, YSB, OCO-2, and OCO-3, assuming that error statistics within each subset are homogeneous. Because this method was originally developed for satellite data, which has wide spatial coverage but infrequent revisit cycles, we applied it directly for satellite observations. We assumed satellite observation errors are spatially uncorrelated and fitted a function (Limited-memory Broyden-Fletcher-Goldfarb-Shanno with Box constraints in R language) to the covariance as a function of separation distance. For ground-based observations, which provide data over a long period, we assumed that observation errors are temporally uncorrelated. Instead of using spatial distance, we fitted a function to the covariance as a function of time steps. The inferred observation error for each dataset was obtained by subtracting the value of the fit at zero separation (distance or time step) from the total covariance. The fitting results for each observation type are shown in Fig. S6. The inferred observation error was multiplied by the square of the standard deviation of departures, and the square of the mean departure was added to correct for bias. Finally, the estimated observation error for each observation was placed on the diagonal of the R matrix.

## 3 Results and discussion

In Sect. 3.1, we compare $CO_2$ emissions from prior and posterior estimates to investigate the spatiotemporal differences following the inversion run. In Sect. 3.2, we assess $CO_2$ enhancement between observations and simulations using prior and posterior emissions to evaluate the effectiveness of the inversion. In Sect. 3.3, we conducted a series of sensitivity analyses to evaluate the robustness of the inversion results. We examined the constraint effects by performing sensitivity tests using all observations, only ground-based data, only OCO-2 data, and only OCO-3 data, along with UR analysis. Additionally, we investigated how the assumptions in the inverse modelling framework affect the inversion outcomes by varying the treatment of background concentrations for ground-based and satellite observations, biogenic fluxes, and the relative uncertainties assigned to prior emissions. A comprehensive discussion accompanies each set of results.

## 3.1 Comparison between prior and posterior emissions

We obtained posterior $CO_2$ emissions over Seoul for December 2021 using Bayesian inverse modelling, incorporating ground-based and satellite observations. Figure 5a compares the average $CO_2$ emissions between the prior and posterior estimates. The mean daytime prior and posterior emissions were 34.12 and 35.63 µmol m$^{-2}$ s$^{-1}$, corresponding to approximately 2.43 and 2.54 million metric tons of $CO_2$ for December 2021 over Seoul, respectively. The average correction from prior to posterior emissions was approximately +4.43%, suggesting a slight increase in posterior emissions, and the prior emissions were slightly underestimated. However, the difference between the domain- and time-averaged prior and posterior emissions was not statistically significant. Similar findings have been reported in previous studies conducted in Tokyo (Ohyama et al., 2023). The reduced chi-squared value for the posterior emissions was 1.38. Although this is slightly higher than the ideal value of 1.0, it still indicates a reasonable representation of prior emissions error and observation error covariance assumptions. Further reducing this value closer to 1 would require assuming larger error metrics.

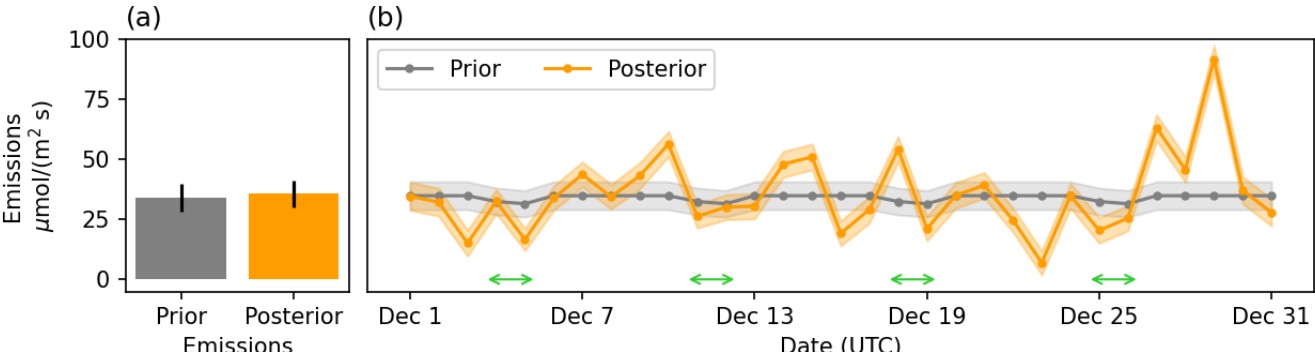

**Figure 5: (a) Comparison of time- and domain-averaged (daytime average) emissions between prior and posterior estimates for December 2021 in Seoul, and (b) time series of domain-averaged daily (daytime average) prior and posterior $CO_2$ emissions, including emission uncertainties. Green arrows indicate weekends.**

The daily time series of temporally resolved prior and posterior emissions averaged over the Seoul domain is shown in Fig. 5b. The prior emissions show relatively low temporal variability, with a standard deviation of 1.28 µmol m$^{-2}$ s$^{-1}$ and a range from 31.5 to 34.86 µmol m$^{-2}$ s$^{-1}$. In contrast, the posterior emissions exhibit substantially greater temporal variability (standard deviation = 16.18 µmol m$^{-2}$ s$^{-1}$, range: 6.95–91.73 µmol m$^{-2}$ s$^{-1}$), capturing realistic fluctuations driven by diverse sources such as traffic, building heating, manufacturing, and energy production. For the prior emissions, the monthly ODIAC data was pre-processed using temporal scaling factors, resulting in slightly higher emissions on weekdays and lower emissions on weekends. The posterior time series did not show a clear distinction between weekday and weekend patterns; however, the average emissions were 38.16 and 28.36 for weekdays and weekends, respectively, with weekday emissions being 1.35 times higher than those on weekends. Posterior emissions fluctuated throughout the month, with a sharp increase from December 27 to 29,

reaching values close to two standard deviations above the monthly mean. This rise may associated with higher heating demand due to lower temperatures and a rebound in traffic activity following the holiday.

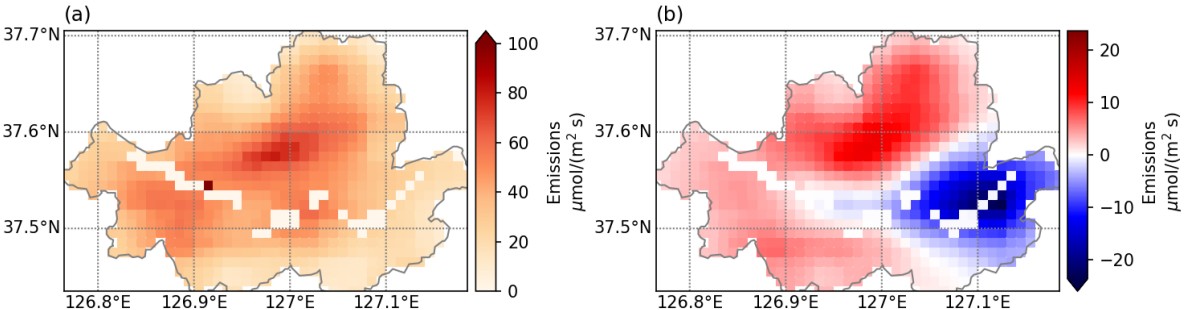

**Figure 6: Averaged CO₂ fluxes over Seoul for daytime over December 2021. (a) Posterior emissions and (b) emission corrections (posterior minus prior) after the inversion run that used five ground sites, OCO-2, and OCO-3 data.**

Figure 6 shows the spatial distribution of posterior $CO_2$ emissions and emission corrections from the inversion using all observation data over Seoul. Compared to the spatial patterns of prior $CO_2$ emissions in Fig. 4a, the posterior emissions in Fig.
6a exhibit a similar distribution, with higher emissions concentrated in central Seoul and lower emissions near the city's boundaries. However, the posterior correction map, obtained by subtracting prior emissions from posterior emissions, reveals spatial variations in emission adjustments ranging from −23.59 to 13.61 µmol m$^{-2}$ s$^{-1}$ (Fig. 6b). Most areas in Seoul experienced either increased or decreased emissions through Bayesian inverse modelling. Notably, emissions in the eastern part of Seoul were strongly corrected in the negative direction, whereas most other regions underwent positive corrections. This suggests
that prior emissions were overestimated in the eastern part of Seoul and underestimated in the other areas. The eastern part of Seoul, including Songpa District, is dominated by residential areas with lower traffic and industrial activity compared to central Seoul. As ODIAC's spatial allocation relies on nighttime light intensity, which can be misleadingly high in residential zones, prior emissions were likely overestimated, resulting in negative corrections in the posterior estimates. In contrast, the central part of Seoul, particularly Jung-gu, exhibited strong positive corrections in the posterior emissions. This area represents the
city's main commercial and business district, characterized by dense traffic networks, high daytime population, and substantial building energy use. As noted by Oda et al. (2018, 2019), ODIAC's spatial disaggregation method does not fully capture emissions from line sources such as the transport sector, which can lead to underestimation in areas with heavy road traffic. Consequently, the posterior optimization increased emissions in Jung-gu to better match the observed concentrations, correcting for the limitations of the prior inventory.

## 3.2 Comparison of CO$_2$ enhancement between observation and simulation

A comparison between observed CO$_2$ enhancements (OBS) and simulated CO$_2$ enhancements (MOD) was conducted to evaluate the performance of the inversion framework and assess how well atmospheric CO$_2$ data constrained emissions over Seoul. Figure 7 shows the changes in the relationship between OBS and MOD from prior to posterior emissions. MOD, based on prior emissions, was significantly lower than OBS, with a slope of 0.39 in Fig. 7a. However, after the inversion, the discrepancies were reduced, and MOD became closer to OBS, with the slope increasing to 0.71. Although observational errors are typically smaller than prior emission uncertainties, which brings the posterior MOD closer to OBS, the posterior slope does not reach 1 because the inversion balances uncertainties from both the observations and the prior emissions. The correlation coefficient between OBS and MOD improved from 0.46 to 0.85, indicating a better agreement after the inversion.

The MAE between OBS and MOD also decreased significantly, from 12.18 to 6.64 ppm, when using posterior emissions data, shown in Fig. 7b. The MAE between observed and modelled $\triangle$CO$_2$ from posterior emissions for the different background configurations were 6.84, 6.64, and 7.20 ppm for GM1, GM2, and GM3, respectively, and 6.76, 6.64, and 6.65 ppm for SM1, SM2, and SM3, confirming that GM2 and SM2 provided the best agreement with the observations. These results indicate that the inversion framework effectively constrained CO$_2$ emissions over Seoul by incorporating information from observational data.

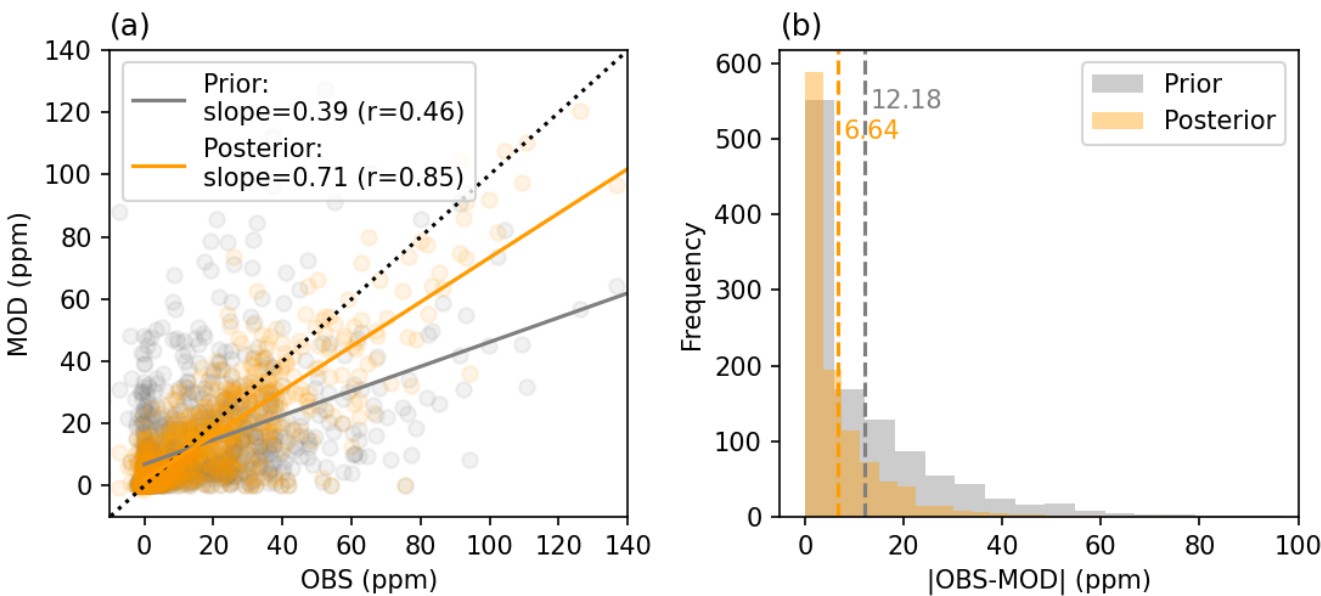

**Figure 7: Comparison of observed CO$_2$ enhancements (OBS) and simulated CO$_2$ enhancements (MOD) using prior (gray) and posterior (orange) emissions during daytime over December 2021. (a) Scatter plots of OBS vs. MOD, with the slope and correlation coefficient (r). (b) Frequency distributions of absolute differences between OBS and MOD (|OBS-MOD|). The dashed lines in (b) indicate the average MAE between OBS and MOD.**

### 3.3 Sensitivity test

#### 3.3.1 Observational network

We evaluated the performance of the inverse model through a sensitivity analysis considering different observational network configurations (Fig. 8). The cases included 1) using all observation data, 2) using only five ground sites, 3) using only OCO-2 data, and 4) using only OCO-3 data. The most substantial reductions in uncertainty were observed when all available observations were used, with an average reduction exceeding 19.2% (Fig. 8a). In this case, the spatial distribution of UR was similar to that obtained using only ground sites (Fig. 8b), as ground-based observations had a greater influence on the constraint

than satellite data because of their continuous temporal coverage throughout the entire month. The spatial difference in UR between using all observations and using only the five ground sites is shown in Fig. S7. When using five ground sites, the UR in posterior emissions was 18.7%. The largest reductions occurred where ground observation sites were concentrated, particularly around OLP (eastern region), SNU (southern region), and surrounding $NST_H$, $NST_L$, and YSB. The reduction in uncertainty extended beyond the observation sites, covering the broader footprint influence range in Fig. 3a. Notably, a

decrease was observed northwest of the observation sites, the upwind region. However, the impact on UR was minimal in western and northern Seoul, where no ground-based observation sites were present.

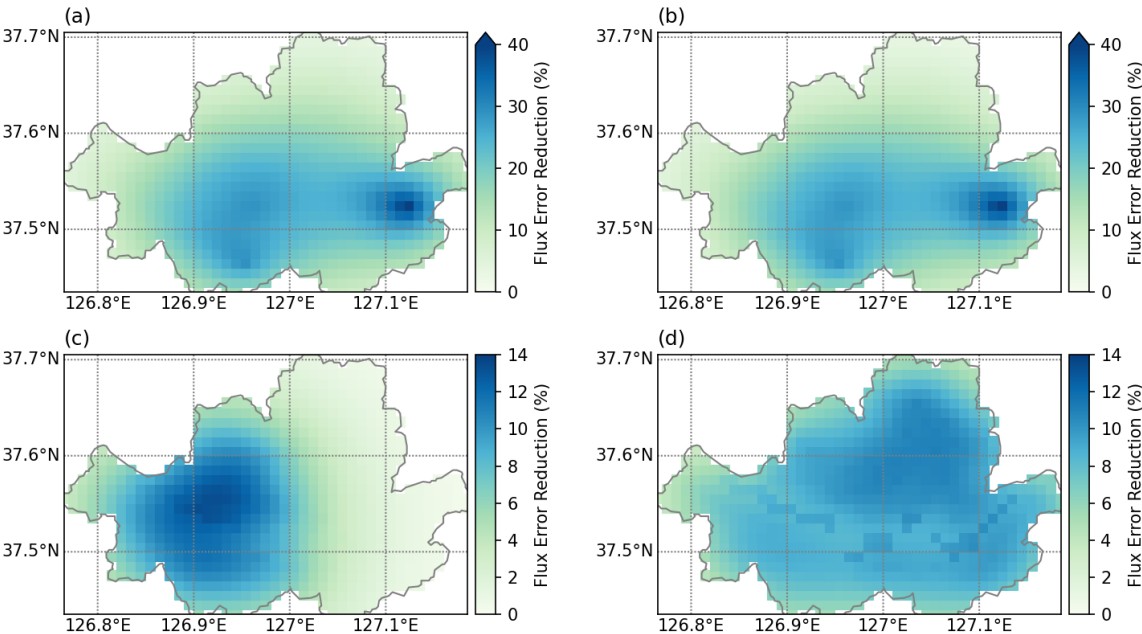

**Figure 8: Daytime percent UR in posterior emissions over Seoul for the inverse analysis using different observation datasets: (a) all observation data from December 2021, (b) only five ground sites from December 2021, (c) only OCO-2 data from December 3–5, 2021, and (d) only OCO-3 data from December 3–5, 2021, when the satellite passed over Seoul. Note that the percent reduction ranges in panels (a), (b), (c), and (d) vary for visualization purposes.**

Because OCO-2 and OCO-3 passed over Seoul on December 4 and 5, 2021, respectively, the UR results for satellite data are focused on December 3–5, highlighting the effect of satellite observations. The OCO-2 data significantly reduced the uncertainty of posterior emissions over western Seoul, with an average reduction of 6% (Fig. 8c). This can be attributed to the satellite's north-to-south overpass, covering the western footprint, an upwind region in Fig. 3b. OCO-2 contributed to UR in the western parts of Seoul, areas not covered by ground-based observations. Most of Seoul experienced UR due to OCO-3, which had 167 soundings across the city (Fig. 8d). The domain-averaged UR from December 3–5 was 8.4%. OCO-3 contributed to UR in the northern region of Seoul, which was not covered by ground-based or OCO-2 observations. The extensive coverage of satellite observations enabled further corrections in areas lacking ground observations, demonstrating the added value of satellite constraints in reducing uncertainty.

The impacts of different observational datasets on the posterior emission estimates are summarized in Fig. S8. Satellite observations influenced the inversion only on the days when OCO-2 or OCO-3 overpassed Seoul, with corrections primarily occurring around December 3–5. Specifically, on December 3–4, satellite-based corrections increased emissions while ground-based observations tended to reduce them, reflecting opposing influences. These differences highlight that the spatial and temporal coverage of observations can lead to distinct patterns of emission corrections across the inversion domain. On December 5, OCO-3 and ground-based data contributed to decreased emissions with nearly consistent magnitudes.

### 3.3.2 Model inputs and assumptions

To evaluate how the assumptions made for components of the inverse modelling framework affect the inversion results, we conducted a series of sensitivity analyses considering background concentrations for ground-based and satellite observations, biogenic fluxes, and the relative uncertainty assigned to prior emissions (Fig. 9). The sensitivity experiments were performed based on a reference configuration, which used the GM2 method for ground-based background estimation, the SM2 method for satellite-based background estimation, included biogenic fluxes, and assumed a 100 % relative uncertainty for prior emissions. Each sensitivity experiment altered only one variable at a time while keeping the others fixed at the reference configuration.

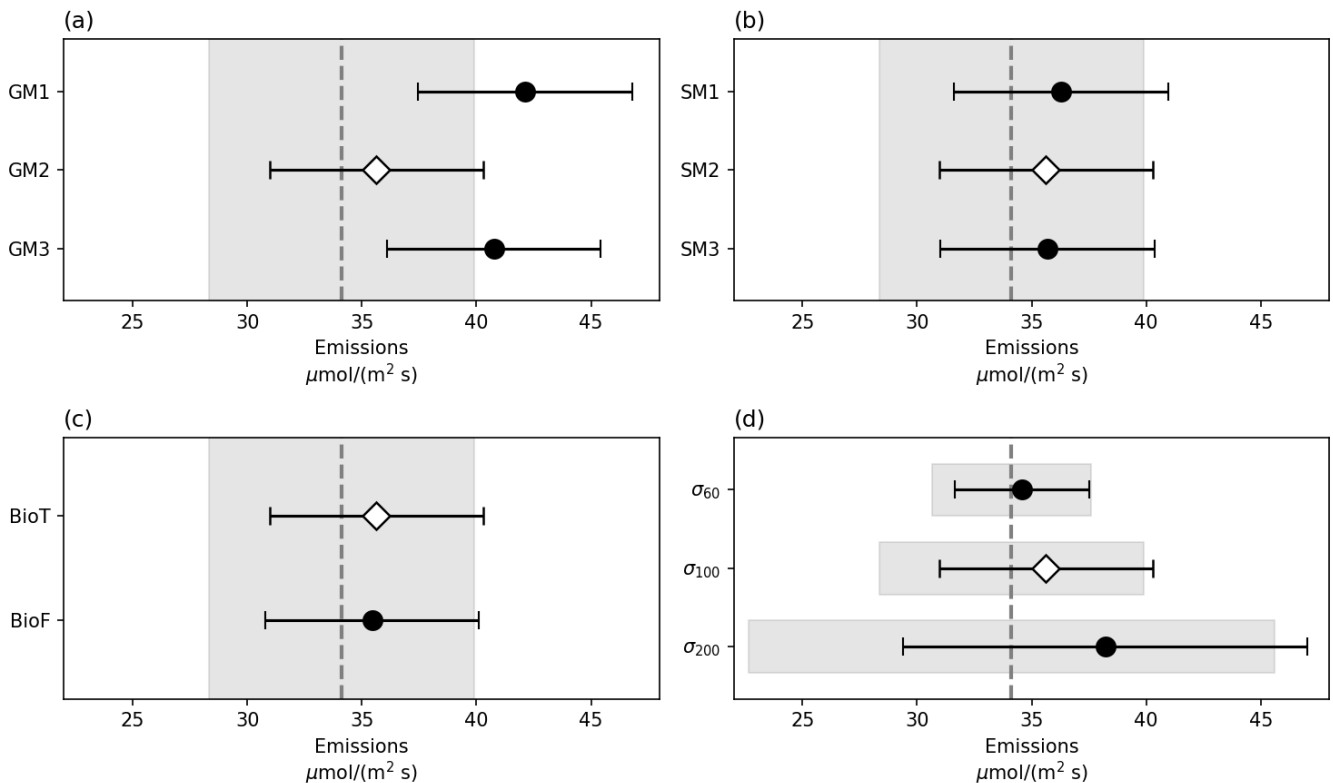

**Figure 9: Sensitivity tests showing changes in posterior CO₂ emissions for (a) ground-based observation background, (b) satellite observation background, (c) with and without biogenic flux, and (d) the relative uncertainty of prior emissions for December 2021 in Seoul. The vertical dashed line and shaded area represent the averaged prior CO₂ emissions and their uncertainty range, respectively. The diamond symbols denote the posterior emissions derived from the reference inverse analysis, while the filled circles indicate the results from sensitivity experiments.**

Regardless of the assumptions, the $CO_2$ emissions increased after the inversion in all experiments. In the ground-based background experiment, the degree of increase in posterior emissions varied substantially depending on the background estimation method (Fig. 9a). When the GM2 method (reference) was used, the posterior emissions increased by 4.43% compared with the prior emissions, while the GM1 and GM3 methods resulted in increases of 23.48% and 19.47%, respectively. This large difference arises from the substantial variation in the background $CO_2$ levels among methods. The mean background concentrations averaged over all ground sites were 443.45 ppm for GM2, whereas GM1 and GM3 produced much lower averages of 436.01 ppm and 437.98 ppm, respectively. Consequently, the derived $\triangle CO_2$ values were larger for GM1 and GM3. During periods when GM2 adjusted the emissions upward, GM1 and GM3 also increased their estimates but to a much larger extent, resulting in substantial overestimation (Fig. S9). Although GM2 was selected as the reference configuration because it yielded the smallest MAE between the posterior-modelled and observed enhancements, the large variation in posterior

emission magnitudes across background estimation methods highlights the critical importance of selecting an appropriate background representation for urban inversions.

For the satellite-based background experiment (Fig. 9b), the differences among background estimation methods were smaller than those for the ground-based cases. Starting from a prior mean of 34.12 μmol m$^{-2}$ s$^{-1}$, the posterior means obtained using SM1, SM2, and SM3 were 36.28, 35.63, and 35.69, respectively. This is because the estimated background concentrations were relatively consistent across the SM1, SM2, and SM3 methods—417.69, 418.63, and 418.24 ppm for OCO-2 and 417.48, 418.64, and 418.72 ppm for OCO-3, respectively. Because satellite $CO_2$ concentrations are generally lower than ground-based values and each satellite observation represents a wide spatial area, the resulting $\triangle CO_2$ is more sensitive to background assumptions, giving the satellite data relatively strong leverage in the inversion (Wu et al., 2018). In this study, however, the background concentrations derived from SM1, SM2, and SM3 were relatively similar, resulting in comparable posterior emissions.

Including biogenic fluxes slightly increased the posterior emissions by 0.17 μmol m$^{-2}$ s$^{-1}$ compared with the case without biogenic fluxes (Fig. 9c). The higher posterior emissions when biogenic fluxes were included can be attributed to daytime photosynthetic $CO_2$ uptake by vegetation. If the observed $CO_2$ concentrations are affected by this uptake, they represent an atmosphere that has already been partially depleted of $CO_2$. Thus, the measured concentrations imply greater fossil-fuel emissions to produce the same observed levels. The resulting increase in $\triangle CO_2$ causes the inversion to adjust emissions upward. However, the overall effect was small, likely because the study period (December) corresponds to the dormant season with limited biogenic activity.

Finally, the sensitivity experiment on the relative uncertainty of prior emissions showed that larger prior uncertainties led to stronger emission adjustments (Fig. 9d). As the relative uncertainty increases, the prior emission covariance matrix becomes larger, effectively giving greater weight to the observational constraints during inversion. The UR were 15.5%, 19.2%, and 23.2% for relative uncertainties of 60%, 100%, and 200%, respectively. These results demonstrate that the selection of appropriate uncertainty values for prior emissions is crucial in determining both the magnitude of posterior emissions and the UR in urban-scale inverse modelling.

## 4 Summary and conclusions

This study developed a Bayesian inverse modelling framework (version 1) to optimize $CO_2$ emissions using both ground- and space-based observations and applied it to Seoul. By incorporating high-resolution (0.01° spatial, 1 h temporal) anthropogenic and biogenic $CO_2$ fluxes, atmospheric $CO_2$ measurements, a Lagrangian transport model, and uncertainty quantification, we improved the accuracy of emission estimates. For Seoul, ground-based ($NST_H$, $NST_L$, OLP, SNU, YSB) and satellite (OCO-2, OCO-3) observations from December 2021 were used to constrain emissions, with WRF-(X)STILT footprints linking emissions to observed $CO_2$ enhancements.

Integrating ground- and satellite-based observations revealed substantial spatiotemporal variations. Posterior emissions exhibited greater variability, capturing both underestimation in most areas and overestimation in eastern Seoul. This combined observational approach not only improved overall emission estimates but also provided detailed insights into spatial and temporal patterns at fine scales, which are critical for tracking when and where $CO_2$ emissions fluctuate and assess the impact of carbon reduction policies over time and space. The inversion highlighted periods of substantial emission increases and identified areas of under- and overestimation, providing information to improve bottom-up emission inventories and offering important policy implications for urban climate action and emissions mitigation. Comparing observed and simulated $CO_2$ enhancements confirmed the effectiveness of the inversion, reducing the MAE by nearly half. Sensitivity tests showed that ground- and space-based data achieved the greatest UR (19.2%), with OCO-2 and OCO-3 providing critical constraints in areas lacking ground-based observations. These results emphasize the complementary role of satellite data, particularly OCO-3's snapshot capability, in enhancing urban $CO_2$ monitoring. We also revealed that assumptions regarding background concentrations, biogenic fluxes, and prior emission uncertainties can meaningfully influence posterior emissions, underscoring the importance of carefully evaluating model inputs.

Despite these improvements, some limitations remain, which future work will address. First, the inversion results depend on the accuracy of the transport model, introducing uncertainties in simulating atmospheric transport. Comparison with observed wind speeds indicates that the model slightly overestimates wind, which could lead to a modest underestimation of posterior $CO_2$ emissions by enhancing simulated transport and dilution. Future improvements to WRF-(X)STILT will aim to reduce discrepancies between modelled and observed meteorological fields through enhanced terrain elevation data and observation nudging. And this study relied solely on daytime observations, and therefore the influence of $CO_2$ concentrations associated with other periods, such as morning and evening rush-hour emissions, was not reflected in the inversion results. Future work will focus on improving the model representation of the nocturnal planetary boundary layer to enable the use of the full diurnal cycle of observations, which would provide better constraints on $CO_2$ emissions. Second, the analysis focused on a 1-month period, which was intentionally selected to evaluate the initial performance of the inversion framework (v1) and to identify areas for improvement. We selected 2021 as it was the first year with complete $CO_2$ measurements from all ground-based sites in Seoul. December was chosen because biogenic influences are minimal and both OCO-2 and OCO-3 passed over Seoul during this month, enabling consistent integration of ground-based and satellite observations. In future work, the inversion

system will be extended to longer temporal periods to capture seasonal variations in emissions and to enable more comprehensive evaluation of the results through comparisons with Korea's national greenhouse gas inventory and previous studies. Third, although the reduced chi-square value has been evaluated, additional validation using independent datasets such as radiocarbon ($\Delta^{14}C$) and flux tower measurements will be conducted.

This approach can be applied to other cities, enabling an independent evaluation of city-reported emissions and identifying discrepancies between reported and observation-constrained estimates. By diagnosing the causes of these differences and progressively reducing them, more robust and transparent emission estimates can be achieved, providing valuable insights for urban mitigation planning and supporting the development of evidence-based climate policies.

**Data Availability**

The prior emissions data from ODIAC were obtained from the Center for Global Environmental Research (https://www.odiac.org/data-product.html). Biogenic $CO_2$ fluxes data from the CASS were downloaded from the Korea Carbon Project website (https://korea-carbon-project.org/map). The WRF model is freely distributed to the scientific community by the National Center for Atmospheric Research (NCAR) and can be accessed at https://www2.mmm.ucar.edu/wrf/users/download/get_source.html. The STILT and X-STILT models are available for installation at https://uataq.github.io/stilt/#/ and https://github.com/uataq/X-STILT, respectively. $CO_2$ measurements from the five ground observation sites used in this study are available upon request from the corresponding author. OCO-2 and OCO-3 satellite data are publicly available from https://ocov2.jpl.nasa.gov/science/oco-2-data-center/ and https://ocov3.jpl.nasa.gov/science/oco-3-data-center/, respectively.

**Author contributions**

SJ guided the research, and SS developed the inverse modelling framework and wrote the manuscript.

**Competing interests**

The authors declare that they have no conflict of interest.

**Acknowledgments**

This work was supported by Korea Environmental Industry & Technology Institute (KEITI) through "Project for developing an observation-based GHG emissions geospatial information map", funded by Korea Ministry of Environment (MOE) (RS-

2023-00232066), and by the Carbon Neutrality Core Technology Development Program (RS-2023-00267529, 2410000450), funded by the Ministry of Trade, Industry & Energy (MOTIE, Korea) and Korea Planning & Evaluation Institute of Industrial Technology (KEIT, Korea).

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
