# Peer review of "Constraining urban fossil fuel CO2 emissions in Seoul using combined ground and satellite observations with Bayesian inverse modelling"

_EGUsphere, 2025_

## Author Response (AR1)

Review for the manuscript "Constraining urban CO2 emissions in Seoul using combined ground and satellite observations with Bayesian inverse modelling"

General comments

This study presents a Bayesian inversion framework that integrates ground-based and satellite observations of atmospheric CO2 to optimize ODIAC's emissions estimates over Seoul. The manuscript is clearly written, and the scientific results are well structured and thoughtfully explained. The work has strong scientific merit, given both the timeliness of the topic and the unique characteristics of Seoul as a valuable testbed—its city size, dense observational network, and active engagement of local government and universities. The inclusion of sensitivity analyses to assess uncertainty enhances the credibility and robustness of the findings.

(answer) Thank you for your positive and encouraging comments. We appreciate your valuable feedback and have revised the manuscript accordingly. In particular, we carefully considered the points you raised and incorporated them into the text. We also expanded the sensitivity analyses to assess how different variables and model assumptions influence the results, and we organized these findings into a new dedicated section to improve clarity and completeness.

The overall impact of the study could be strengthened with a few additional analyses or clarifications. In particular, it would be useful to (1) relate daily fluctuations in posterior CO2 emissions to specific events or emission cycles in Seoul during the study period, and (2) provide possible explanations for why ODIAC may systematically underestimate or overestimate emissions in different parts of the city. Addressing these points would add depth to the interpretation and broaden the study's relevance. Further details on these suggestions are provided below.

(answer) Thank you for these constructive suggestions. In Section 3.1, we have added additional explanations addressing both points: 1) how daily fluctuations in posterior $CO_2$ emissions relate to specific events and emission cycles, and 2) why ODIAC may systematically underestimate or overestimate emissions in different parts of Seoul.

Specific Comments

Line 15: The sentence "Additionally, the mean absolute error was reduced, improving the agreement between simulated and observed CO2 enhancements." is somewhat redundant, as the inversion framework is designed to reduce the mismatch between observed and simulated CO2 enhancements. I recommend either removing the sentence or strengthening it by providing quantitative details (e.g., "the mean absolute error decreased by X%, indicating improved agreement"). This would make the point more informative and impactful.

(answer) We agree that providing quantitative information about the mean absolute error would make the statement more informative. Accordingly, we revised lines 16–17 to include the reduction rate of the mean absolute error.

Graphical abstract: Consider adding labels or legends to the three maps within the Footprint box. From context, I assume they represent footprints for ground, OCO-2, and OCO-3 observations, but making this explicit would help readers quickly interpret the figure.

(answer) To make the footprints immediately interpretable, we added labels ("Ground," "OCO-2," and "OCO-3") above each footprint map. For consistency, we also added the same labels to the corresponding observation plots.

Line 41: There appears to be a minor typo in the reference to (40 Cities, 2022)—please double-check the citation format.

(answer) We updated the reference for C40 Cities (2022) to conform to the APC journal citation format, and we rechecked all references to ensure they follow the journal's required style.

Lines 46–57: It would strengthen the discussion to highlight the importance of self-reported emissions inventories by global cities. For most cities, these self-constructed inventories form the basis of climate action planning and emissions mitigation policies. For example, the Global Protocol for Community-Scale Greenhouse Gas Emission Inventories (GPC) provides the standard framework adopted by many C40 cities. Yet, discrepancies have been reported between GPC-based inventories and bottom-up datasets such as EDGAR and ODIAC. Presenting this context would help situate the study within the broader policy landscape. In particular, framing the comparison as simply "bottom-up vs. top-down" may be too simplified. I suggest expanding this section to include how selfreported inventories are currently used by cities, and how ground & satellite-based top-down approaches can provide complementary value for monitoring, validation, and policy assessment.

(answer) Following the recommendation, we expanded Lines 48–64 to describe the role of self-reported emission inventories, particularly those developed using the GPC framework widely adopted by C40 cities. Additionally, to address the reviewer's point regarding the broader policy context, we added text in the conclusion (Lines 602–605) explaining how top-down emission estimates derived from inverse modelling can complement self-reported inventories.

Line 133: Please clarify whether the reported altitude values are given above ground level or above sea level.

(answer) We have clarified this point in the revised manuscript (Lines 150–151).

Figure 1: This is a clear and effective figure. To further enhance it, I suggest including altitude information either directly on the map or in the caption. In addition, it may be helpful to add a box or inset indicating the "background" domain, so readers can easily visualize its scale relative to the city.

(answer) We added information about the typical elevation of Seoul to the Fig. 1 caption. In addition, we marked the satellite background concentration domain with a red box in the inset map located in the upper-left corner.

Figure 2: This is another strong figure. It would be particularly insightful to add a panel showing the daytime variability of ground-based $CO_2$ mixing ratios (e.g., 10–16 LT) at each station, plotted as hour versus CO2 concentration. Such a panel would highlight how surface $CO_2$ concentrations evolve during the day in a megacity like Seoul. An accompanying paragraph in the results or discussion, describing the observed daytime variability and its relationship to planetary boundary layer dynamics and diurnal emission cycles, would add depth. In panel (b), you might also consider marking the five ground stations with yellow dots so that readers can immediately connect the measurements shown in panel (a) with their geographic locations.

(answer) Following the reviewer's suggestion, we added a new panel (c) in Fig. 2 that presents the diurnal cycle of $CO_2$ concentrations at each ground station, with daytime highlighted. We also incorporated a description of this

daytime variability—relating it to boundary layer dynamics and diurnal emission patterns—into the manuscript (Lines 159–165). In addition, we marked the ground observation sites on the OCO-2 spatial distribution map to help readers easily connect the surface measurements with their locations.

Lines 151–152: The statement "In both datasets, the XCO2 values were higher in the southern part of Seoul compared to the northern part" would benefit from some explanation. Could you suggest possible reasons or hypotheses for this observed north–south gradient (e.g., spatial patterns in emissions or transport influences)? Providing context here would strengthen the interpretation.

(answer) We added an explanation in Lines 188–191 describing possible causes of the north–south gradient observed in the satellite $XCO_2$ data.

Lines 164–166: The choice of using 24-hour moving 5th percentile values for background determination may need additional justification, especially since the analysis itself focuses only on 10–16 LT. What assumptions underlie this approach? Including a rationale—or, if possible, a sensitivity analysis using alternative background definitions—would increase the robustness of the results.

(answer) We summarized previous studies on defining background concentrations for ground-based observations and selected three approaches applicable to Seoul (Lines 197–213). Among these, we chose the 24-hour moving 5th percentile method because it produced the lowest mean absolute error (MAE) between observed and modelled $\Delta CO_2$ based on the posterior emissions (Lines 467–469). In addition, we added a new subsection (3.3.2 Model inputs and assumptions) presenting a sensitivity analysis that illustrates how the results vary depending on the background definition used for ground sites.

Lines 166–167: A similar point applies to the use of the median value within a ~500,000 km² box. Given that Seoul itself is ~600 km², the chosen background domain is quite large. More explanation is needed as to why this choice is appropriate, and ideally, some assessment of its implications. Since background definition is widely recognized as a challenge in urban inversion studies, adding further description or sensitivity testing here would significantly strengthen the scientific merit of the work.

(answer) For the satellite background concentration, we added a summary of relevant previous studies and described the three approaches tested for application to the Seoul inversion (Lines 214–244). Among these, the method using non-urban areas within the previously defined ~500,000 $km^2$ box yielded the lowest MAE (Lines 467–469), supporting the appropriateness of this domain. We also included a sensitivity analysis of the satellite background definitions in Section 3.3.2.

As noted above, including this background box in Figure 1 (as an inset) would also help readers visualize its relative scale.

(answer) The corresponding figure has been updated accordingly.

Line 186: In the sentence "For WRF-STILT, one thousand air particles were released from each observation site and tracked backward in time for 24 h (Fig. 3a)," please specify how frequently WRF-STILT was run for each site (e.g., hourly? or once per day?). This detail will improve reproducibility and interpretation.

(answer) We provided a detailed description of the WRF-STILT simulations in Lines 270–277, specifying that the model was run hourly to produce hourly footprints. Additional details on particle release height, vertical mixing, and footprint computation were also included to improve reproducibility and interpretation.

Figure 3: Okay, now I see each panel is for ground, OCO-2, and OCO-3! To further improve clarity, I recommend adding the representative times for each panel. For example, panel (a) could represent average daytime conditions during December 2021, panel (b) 13:00 LT on December 4, 2021, and panel (c) 11:00 LT on December 5, 2021. Explicitly including this information would make the figure more self-explanatory.

(answer) We added the representative times to the Fig. 3 caption, specifying the periods or local times for ground, OCO-2, and OCO-3 observations to make the figure more self-explanatory.

Lines 220–221: Returning to my earlier comment on Figure 2, while I understand that diurnal CO2 variability is not the primary focus of this study, a brief discussion of hourly CO2 changes would add valuable context. Insights

derived from ground observations—such as the influence of traffic, heating, or boundary layer growth—would broaden both scientific and policy relevance of the results, connecting the inversion framework to real-world urban dynamics.

(answer) Regarding the comment on hourly $CO_2$ variability, we refer the reviewer to our earlier response: Following the reviewer's suggestion, we added a new panel (c) in Fig. 2 that presents the diurnal cycle of $CO_2$ concentrations at each ground station, with daytime highlighted. We also incorporated a description of this daytime variability—relating it to boundary layer dynamics and diurnal emission patterns—into the manuscript (Lines 159–165). We also added a description of the daily emission patterns in Lines 313–315.

Figure 4. In panel (a), there appears to be a dark red pixel which I would assume as the location of a power plant, correct? If so, it would be nice to have a label identifying the power plant on the figure. In addition, I recommend including a note in the caption describing how the power plant location is represented in the original ODIAC dataset. This additional detail would not only improve figure readability but also provide useful context for future studies that make use of ODIAC in the Seoul region.

(answer) We added a label for the power plant in Fig. 4(a) and included a note in the caption indicating that the power plant location was adjusted from the original ODIAC dataset.

Line 312: It would be very helpful to translate the reported emissions into mass units on a monthly scale (e.g., million metric tons of CO2 per month). How does your estimate compare to previous studies that estimated Seoul's CO2 emissions using ground and/or satellite observations? Presenting the results in these terms would provide readers with an intuitive reference point and allow for easier comparison with other inventories or policy-relevant metrics.

(answer) We converted the emission values originally expressed in $\mu mol\ m^{-2}\ s^{-1}$ into monthly totals in units of million metric tons of $CO_2$ per month and added this information to the manuscript (Lines 407–409). Unfortunately, a direct comparison with previous estimates is difficult because the Seoul Metropolitan Government provides official $CO_2$ emissions only on an annual basis, and existing studies that estimate Seoul's greenhouse gas emissions use different time periods that do not align with our study.

Figure 5. Consider adding vertical lines or shaded area to indicate the weekends in panel b.

(answer) We have updated the figure accordingly by adding arrows to indicate the weekends, along with a brief explanation (Lines 426–430).

Line 327-328 The phrase "reflecting a more realistic pattern" could be made more objective. For example, you could refer to temporal variability (1sigma, min, max) from observation-constrained vs. ODIAC/TIMES.

(answer) We have revised the sentences to present the comparison more objectively using statistical measures (Lines 423–426).

Lines 328-329: As currently written, the text suggests that posterior emissions fluctuate only until December 22, whereas they appear to vary throughout the entire month. If the intent is to emphasize the increase observed during December 27-29, I recommend rephrasing to state that emissions during these days exceed the typical rage of variability (for example, outside of two standard deviations of the monthly fluctuations).

(answer) We revised the text in Lines 430–432 to indicate that posterior emissions fluctuated throughout the month and highlighted that values from December 27–29 exceeded the typical range of variability based on the standard deviation. We also added a discussion of potential reasons for this increase, including higher heating demand and a rebound in traffic activity.

Lines 335-338: The negative correction (i.e., reduced posterior emissions) seems to be centered around the OLP site. It would strengthen the discussion to include a short description of the characteristics of the eastern part of Seoul compared with other areas of the city. For instance, is this region more residential versus commercial, or less of a business district? Providing this context could help explain why ODIAC may underestimate emissions in this particular area. More broadly, it would be useful to reflect on the conditions under which nighttime light imagery—used for ODIAC's spatial allocation—tends to perform well, and when it may introduce biases. This would add depth to the interpretation and provide guidance for future applications of ODIAC in urban settings.

(answer) We expanded the discussion in Section 3.1 to explain the spatial context behind the correction patterns (Lines 445–454). We describe that the eastern part of Seoul is primarily residential with lower traffic and industrial activity, leading ODIAC's nighttime-light-based allocation to overestimate emissions and resulting in negative posterior corrections. Conversely, we added a brief explanation of why central districts such as Jung-gu show positive corrections, noting ODIAC's known limitations in representing line-source emissions from dense traffic.

Figure 7. This is more of a question than a direct comment, as I am not an expert in inversion frameworks. Why is it that the slope between modeled (MOD) and observed (OBS) values in the posterior cannot be optimized to 1 (i.e., perfectly fitting MOD to OBS on the average scale)? Is this related to the configuration of the error covariance matrices—for example, if the uncertainty assigned to prior emissions is relatively strict compared to the observational uncertainties? If so, does this imply that if the inversion were configured to allow a perfect match between MOD and OBS, the resulting posterior emissions would be higher than the current estimates? Including a few sentences in the text to briefly address this point would be helpful for readers who may share this question.

(answer) We have added a brief explanation to clarify why the posterior slope does not reach 1 (Lines 461–464). Specifically, we note that the inversion balances uncertainties from both the observations and the prior emissions. Although observational uncertainties are generally smaller and therefore pull the posterior MOD closer to the OBS, the constraints imposed by the prior prevent the posterior slope from fully reaching 1. Consequently, achieving a perfect 1:1 fit would result in higher posterior emissions than those currently estimated.

Figure 8. There's typo in the caption. (b) appears twice in the caption.

(answer) We have corrected the duplicated panel label in the caption of Fig. 8.

Section 3.3: It would be very informative to see how the emission estimates differ across the various sensitivity experiments (all data vs. ground-only vs. OCO-2 vs. OCO-3). One way to highlight this would be to include an additional figure, similar in style to Figure 5, that shows how the average emissions and their fluctuations change under each experiment. Such a comparison would make the relative contributions of ground-based and satellite observations to the inversion more straightforward and would help readers better understand the value added by

each dataset.

(answer) We have incorporated additional analysis comparing the posterior emissions across the different sensitivity experiments, and the corresponding description has been added to Lines 509–514 along with a new summary figure (Fig. S8).

Lines 397-399: The sentence "Although the averaged CO2 emissions difference between prior and posterior estimates was relatively small (4.43% increase), the inversion revealed significant spatiotemporal variations" may unintentionally understate the impact of the study's findings. I suggest reframing this point to highlight the value of integrating ground- and satellite-based observations- Beyond improving overall emission estimates, this approach provides insights into spatial and temporal variations at fine scales, which carry important policy implications for urban climate action and emissions mitigation.

(answer) We revised the text to more clearly emphasize the value of integrating ground- and satellite-based observations. The updated sentences now highlight how this combined approach enhances both the accuracy of overall emission estimates and the understanding of fine-scale spatial and temporal emission patterns, which carry important policy relevance for urban climate mitigation (Lines 572–578).

Data Availability: The current section appears to be missing on the availability of ground-based observations. Please include a citation for the data source and indicate whether these observations are publicly available (and if so, how they can be accessed).

(answer) We have added information on the availability of the ground-based $CO_2$ observations in the Data Availability section, noting that the measurements from the five sites can be obtained upon request from the corresponding author (Lines 613–614).

Review of "Constraining urban CO2 emissions in Seoul using combined ground and satellite observations with Bayesian inverse modelling" by Sim and Jeong, 2025.

This study presents an atmospheric inverse modelling framework to estimate fossil fuel CO2 emissions in Seoul megacity. The study is timely and well within the scope of ACP, addressing the critical need for independent verification of urban CO2 emissions. The main novelty is the combined assimilation of both in situ and satellite observations within an urban-scale inversion. The framework is clearly described and general enough to be of value for the design of other urban inversion systems. The inversion is applied for the month of December 2021, finding a slight increase in total CO2 emissions compared to the prior ODIAC inventory. While the study is generally well-executed, several aspects would benefit from further clarification or discussion. In particular, the suitability of the background definition, the potential influence of other model errors, and a more detailed interpretation of the posterior flux estimates would provide stronger support for the conclusions. Nonetheless, the study demonstrates sufficient merit as a proof-of-concept of the inversion framework. I therefore recommend publication after minor revisions, provided the authors address the following comments.

(answer) We sincerely thank the reviewer for the positive and constructive assessment of our manuscript. We appreciate the recognition of the novelty and value of our urban-scale inversion framework, as well as the encouraging recommendation for publication pending minor revisions. In response to the reviewer's comments, we have revised the manuscript to address the key points raised. Specifically, we have:

1. Improved the discussion on the suitability of the background definition by providing additional justification for the selected approach and expanding the sensivitiy analysis to comapre with alternative background representations.

2. Assessed the potential influence of biogenic fluxes and prior emission uncertainties, and clarified how these factors affect the resulting posterior flux estimates.

3. Enhanced the interpretation of the posterior emission results, including a more detailed explanation of the inferred spatiotemporal variability in fossil fuel $CO_2$ emissons.

We believe these revisions have significantly improved the clarity and robustness of the manuscript.

General comments:

Background definition: The definition of the background for both in situ and satellite observations is a critical component of the urban-scale inversion. At present, no clear justification for the chosen background definitions is provided, making it difficult to assess their validity. For the in situ sites, the background is defined as the 5th percentile within a 24-hour moving window. Could the influence of biospheric uptake within the domain, or periods when the wind direction shifts within the window, bias this estimate compared to the true background? For satellite observations, the background value is defined as the daily median XCO2 over non-urban areas within a 500,000 km2 domain centered on Seoul. It is not immediately obvious whether this definition is appropriate. Given the spatial pattern of the posterior flux corrections, it seems plausible that these patterns may, at least in part, reflect biases in the background definition. A short analysis demonstrating why the chosen background is suitable, or even a sensitivity test using an alternative background, would help in evaluating the robustness of the results.

(answer) We summarized previous studies on defining background concentrations for ground-based observations and selected three approaches applicable to Seoul (Lines 197–213). Among these, we chose the 24-hour moving 5th percentile method because it produced the lowest mean absolute error (MAE) between observed and modelled $\Delta CO_2$ based on the posterior emissions (Lines 467–469). For the satellite background concentration, we added a summary of relevant previous studies and described the three approaches tested for application to the Seoul inversion (Lines 214–244). Among these, the method using non-urban areas within the previously defined ~500,000 km$^2$ box yielded the lowest MAE (Lines 467–469), supporting the appropriateness of this domain. To address the reviewer's comment, we added a new subsection (3.3.2 Model inputs and assumptions) that presents a sensitivity analysis assessing how the inversion results vary depending on the chosen background definitions for both ground and satellite observations.

Sensitivity tests: The overall analysis of inversion model performance is somewhat limited. The robustness of the inversion results could be improved by additional sensitivity tests, especially including variations in background concentrations, the influence of biospheric fluxes, prior uncertainties, and transport model uncertainties. I do not view a full suite of new experiments as strictly necessary for publication, but at a minimum the manuscript should include more explicit acknowledgement and discussion of how these potential sources of error could affect the

inversion results. Without these sensitivity tests, the strength of the conclusions is limited and should not be overstated.

(answer) We agree that additional analysis of potential error sources can improve the robustness of the inversion results. In response, we have expanded Section 3.3.2 to include sensitivity tests examining 1) alternative background concentration definitions for both in situ and satellite observations, 2) inversions with and without biospheric fluxes, and 3) different assumptions regarding prior emission uncertainties. These additions provide a clearer assessment of how these factors influence the posterior estimates and strengthen the discussion of potential limitations.

Study period: The study focuses only on the month of December 2021. The rationale for selecting such a short study period is not sufficiently justified in the current manuscript. The short study period limits the strength of the conclusions and frames the work more as a proof-of-concept.

(answer) We have clarified the rationale for selecting December 2021 as the study period and for focusing on a single month. Specifically, we added explanations in Lines 134–139 and 593–600 describing why December 2021 was chosen and why the initial implementation of the inversion framework (v1) was applied to a 1-month period.

Interpretation and policy relevance: The inversion finds a slight increase in net emissions relative to the prior ODIAC inventory, along with an interesting spatial pattern of flux corrections. The manuscript would benefit from a brief discussion of plausible emission sources, processes, or model errors that could explain these differences. Without such context, it is difficult to conclude whether the posterior flux estimate represents a real emissions signal or could instead reflect model biases e.g., a background or transport model error. It would also be useful to expand the conclusions to discuss what the results could mean for current mitigation efforts in Seoul.

(answer) We expanded the discussion of the spatiotemporal variability in the posterior emissions in Section 3.1 to address plausible emission sources and the potential systematic influences of the prior inventory. This addition provides clearer context for interpreting the observed differences between the prior and posterior fluxes. Furthermore, we added a brief discussion in the Conclusions (Lines 572–578) on how the inferred emission patterns could inform and support ongoing urban climate mitigation efforts in Seoul.

Specific comments:

Line 126-129: Include an explanation for why only the month of December 2021 was selected for analysis.

(answer) As noted in our earlier response, we have added an explanation clarifying why December 2021 was selected and why the inversion was conducted for a single month (Lines 134–139 and 593–600).

Line 147-152: A general description of the typical availability of OCO-2 and OCO-3 observations over Seoul (or urban areas in general) would add useful context. For instance, is December 2021 representative in terms of data coverage, or was this a favorable period for high-quality retrievals? Since the study emphasizes the utility of satellite observations and the potential extension of the framework to other urban areas, providing this information would help readers assess how broadly the approach might be applied.

(answer) We added an explanation of the typical availability of OCO-2 and OCO-3 observations over Seoul, along with a clarification of why December 2021 was selected for this study. These additions in Lines 175–185 provide readers with context on the representativeness of the satellite data coverage and help illustrate how satellite observations can be effectively used in urban inverse modelling applications.

Line 164-167: A thorough explanation and justification for the background is required.

(answer) As noted in our earlier response, we have added a detailed explanation and justification for the selected background concentration approach in the manuscript.

Line 170-191: Some aspects of the transport model configuration need clarification. In the case of WRF-STILT: Are the particles released at the same height as the measurement inlet? Is 24 hours sufficient for particles to leave the modeled domain? What is the vertical output layer of the footprints (surface layer, PBL, or other)? In the case of WRF-XSTILT: How are the averaging kernel and a priori profile applied? What happens above 6000 m a.g.l.?

(answer) All requested clarifications for both WRF-STILT and WRF-XSTILT have been added to the manuscript (Lines 270–285). Specifically, we now describe the particle release height, the adequacy of the 24-hour backward

period, the vertical layer used for footprint calculations, the application of the averaging kernel and a priori profile, and the treatment of air parcels above 6000 m a.g.l.

Line 170-191: How do modeled meteorological fields compare with observations? Even small biases (e.g., wind speed or planetary boundary layer systematically higher/lower) could strongly influence the inversion results. This information appears in the supplement, but this should be discussed in the main text.

(answer) We have summarized the comparison between modelled and observed meteorological fields—previously provided only in the supplement—and incorporated it into the main text (Lines 262–269). We also added a brief discussion in the conclusion (Lines 586–590) describing how biases in wind speed could affect inversion results and outlining potential approaches in future work to reduce these biases.

Line 223: "the daytime (1-7 UTC) emissions for December 2021 serve as the state vector". Throughout the text, it would be worth acknowledging the full diurnal cycle of fluxes, and some of the limitations of the current approach for constraining net CO2 For example, morning and evening commuter rush hours are not included in the state vector and therefore not optimized. To what extent does this reduce the ability to constrain net CO2 emissions?

(answer) The state vector represents emissions for the full month of December 2021, while only daytime observations are used. These daytime observations are linked to the corresponding 24-hour backward footprints, which means that each observation constrains the emissions from the preceding 24 hours. Although the inversion framework therefore adjusts emissions across all hours, only daytime posterior emissions are presented in the results section. To avoid confusion, we removed the earlier ambiguous wording and revised Lines 316–317 accordingly. Additionally, we added a discussion in the conclusion (Lines 590–593) describing the limitations associated with using daytime-only observations—particularly for morning and evening rush-hour emissions—and outlining plans to address these issues in future work.

Technical corrections:

(answer) For the technical corrections, we provide concise responses that indicate the relevant revision locations

Line 1: Consider revising "urban CO2 emissions" to "urban fossil fuel CO2 emissions" to avoid ambiguity, since the current phrasing could also include biospheric fluxes. Line 1

Line 8: Define on first use: "CO2" to "carbon dioxide (CO2)". Line 10

Line 8: "the transport model" to "a transport model". Line 10

Line 10: Again, would be useful to specify fossil fuel CO2. Line 12

Line 15: "mean absolute error was reduced" to "mean absolute error between simulated and observed CO2 enhancements was reduced". Lines 16–17

Line 17: "The most substantial reductions in uncertainties (19.2%) were observed when all available observations were used". This goes without saying. It would be more insightful to report the uncertainty reduction for the different observational network configurations in the abstract, so that the contribution from both in situ and satellite observations is clear at the outset. Lines 18–20

Line 55: Drop "the" before "large point sources". Line 62

Line 63-64: "the uncertainty reduction" to "uncertainty reductions". Line 71

Line 92: Suggest writing "0.01° (approximately 1 km)" for easier interpretation. Line 99

Line 114: I recommend replacing the italic bold "Error reduction" in the equation with a defined variable e.g., "The uncertainty reduction (UR) is defined as…". Lines 119–121 and across the entire text.

Line 149: "volume mixing ratio" to "mole fraction". Line 177

Figure 3: Can the unit also be displayed on the colorbar. Also make sure that all colormaps used in the paper are colorblind-friendly. The footprint unit has been added to the colorbar in Fig. 3. We also verified all colormaps using a color-blindness simulator and confirmed that the visual information remains distinguishable, ensuring that all colormaps used in the manuscript are colorblind-friendly.

Line 223: Ensure consistency for reported times, either use a single time zone throughout the text or provide both UTC and KST. We have revised the manuscript to report all times in Korea Standard Time (KST) for consistency.

Line 264-266: Phrasing is incorrect "the time/distance at which errors in the prior emissions are considered

uncorrelated". Technically, correlation at the e-folding correlation lengths is non-zero. Lines 357–360

Figure 7: Please add units to Fig. 7b (ppm?). We have added units to Fig 7b.

Figure 8: It is very hard to see any difference between Fig. 8a and Fig. 8b. In response to the reviewer's comment, we added a new figure (Fig. S7) that illustrates the spatial differences in uncertainty reduction between the all-observation case and the ground-only case, providing a clearer comparison.

Line 420: The provided URL https://db.cger.nies.go.jp/dataset/ODIAC does not appear to be active. Line 609

Line 499: "IPCC: GLOBAL WARMING OF 1.5°C an IPCC special report on the impacts of global, Ipcc, 2018". Incorrectly formatted reference. We updated the reference for IPCC (2018) to conform to the APC journal citation format, and we rechecked all references to ensure they follow the journal's required style.